# No Pixel Left Behind: A Detail-Preserving Architecture for Robust High-Resolution AI-Generated Image Detection

**Lianrui Mu, Haoji Hu,**\* **Xingze Zou, Jianhong Bai, Jiaqi Hu**
Zhejiang University
`{mulianrui,haoji_hu,zeezou,jianhongbai,jiaqi_hu}@zju.edu.cn`

## Abstract

The rapid growth of high-resolution, meticulously crafted AI-generated images poses a significant challenge to existing detection methods, which are often trained and evaluated on low-resolution, automatically generated datasets that do not align with the complexities of high-resolution scenarios. A common practice is to resize or center-crop high-resolution images to fit standard network inputs. However, without full coverage of all pixels, such strategies risk either obscuring subtle, high-frequency artifacts or discarding information from uncovered regions, leading to input information loss. In this paper, we introduce the **Hi**gh-Resolution **D**etail-**A**ggregation Network (**HiDA-Net**), a novel framework that ensures no pixel is left behind. We use the *Feature Aggregation Module* (FAM), which fuses features from multiple full-resolution local tiles with a down-sampled global view of the image. These local features are aggregated and fused with global representations for final prediction, ensuring that native-resolution details are preserved and utilized for detection. To enhance robustness against challenges such as localized AI manipulations and compression, we introduce *Token-wise Forgery Localization* (TFL) module for fine-grained spatial sensitivity and *JPEG Quality Factor Estimation* (QFE) module to disentangle generative artifacts from compression noise explicitly. Furthermore, to facilitate future research, we introduce **HiRes-50K**, a new challenging benchmark consisting of **50,568** images with up to **64 megapixels**. Extensive experiments show that HiDA-Net achieves state-of-the-art, increasing accuracy by over **13%** on the challenging Chameleon dataset and **8%** on our HiRes-50K.

## 1 Introduction

The rapid advancement of AI-generated image (AIGI) technologies, particularly diffusion models proposed in recent works (Sohl-Dickstein et al., 2015; Ho et al., 2020; Dhariwal & Nichol, 2021; Podell et al., 2023; Rombach et al., 2022), has led to a surge in the generation and sharing of hyper-realistic, high-resolution images online. Unlike the outputs of early generative models (Mirza & Osindero, 2014; Karras et al., 2017), generated images on social platforms are often carefully selected, edited, or even upscaled by users (Saharia et al., 2022), making them nearly indistinguishable from real photographs to the human eye (Kamali et al., 2025). This new reality poses significant risks to information authenticity (Ferreira et al., 2020), societal trust, and copyright protection (Ren et al., 2024), making the development of robust detectors for high-resolution generated images a priority.

Despite significant strides in AIGI detection (Wang et al., 2020; Frank et al., 2020; Ojha et al., 2023), the generalization ability of existing methods has notably degraded on modern, high-resolution benchmarks like the Chameleon dataset (Yan et al., 2024a). We attribute the performance collapse to two key challenges: **Input Degradation** and **Limited Generalization**.

**Input Degradation.** A primary cause for the failure of current detectors on high-resolution images is an architectural bottleneck. Most frameworks resize large inputs to a fixed, low resolution (e.g., $224 \times 224$) to fit standard backbones (Ojha et al., 2023; Tan et al., 2024a; Liu et al., 2024; Tan et al.,

---

\*Corresponding author: Haoji Hu.

2025). As shown in Sec. 3, resizing introduces a strong low-pass effect, irreversibly erasing the subtle, high-frequency fingerprints that are most indicative of AI-generated artifacts. While some methods attempt to mitigate this by cropping limited regions, such as TextureCrop's region selection approach (Konstantinidou et al., 2025) or SAFE's center-cropping (Li et al., 2024), they only analyze a few selected regions. This partial analysis discards potentially crucial evidence from the rest of the image. We argue that a truly comprehensive analysis requires systematically examining the entire image at its native resolution to ensure no detail is overlooked.

**Limited Generalization.** As research (Zheng et al., 2024) has shown, models can learn "shortcuts" by overfitting to dataset-specific cues, such as the generation source model or generation prompts, rather than universal synthetic artifacts. This problem is severely exacerbated by mismatched JPEG compression histories between real and fake images, which teaches the model to become a compression detector rather than a synthesis detector (Grommelt et al., 2025). While naive JPEG augmentation offers partial relief, it often fails to generalize to unseen compression levels. Moreover, the rise of localized forgeries like AI-driven inpainting demands that detectors possess fine-grained spatial awareness to identify manipulated regions within otherwise authentic high-resolution images (Chen et al., 2024), posing a significant challenge for models trained solely on datasets with fully synthesized images.

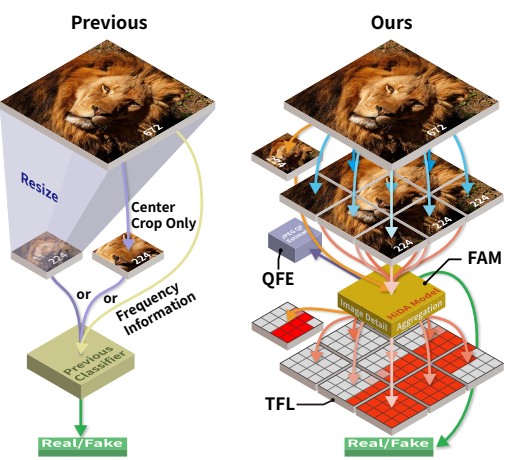

Figure 1: **Comparison between our model and previous approaches.** Many existing methods either resize high-resolution images to fit a visual backbone or crop only the central region. In contrast, our method processes full-resolution tiles covering the entire image, preserving fine-grained details and utilizing all pixel information for more accurate detection on high-resolution images.

To address these multifaceted challenges, we introduce the **Hi**gh-Resolution **D**etail-**A**ggregation Network (HiDA-Net), a framework designed for comprehensive and robust detection of high-resolution AI-generated images. As illustrated in Fig. 1, HiDA-Net avoids input downsampling by processing the entire image as a series of full-coverage, native-resolution tiles. It utilizes a novel **Feature Aggregation Module (FAM)** to fuse features from these local tiles with a global contextual view, ensuring no pixel is left behind. To combat shortcut learning and enhance generalization, we incorporate two extra training tasks. **Token-wise Forgery Localization (TFL)** endows the model with fine-grained spatial awareness to pinpoint manipulated regions, making it robust against localized forgeries like inpainting. And the **JPEG Quality Factor Estimation (QFE)** module, utilizing the preserved pristine JPEG artifacts within each crop tiles, forces the model to disentangle generative traces from compression noise, enhancing the robustness facing JPEG compression. To facilitate rigorous and realistic evaluation, we also present **HiRes-50K**, a new challenging benchmark of *50,568* high-resolution images (up to 64 megapixels) with carefully aligned compression distributions and image sizes. Our main contributions are:

- **A Novel Detail-Preserving Architecture:** We propose **HiDA-Net**, a network that processes full-coverage, native-resolution tiles to prevent information loss. Its key components: the *Feature Aggregation Module (FAM)*, *Token-wise Forgery Localization (TFL)*, and *JPEG Quality Factor Estimation (QFE)*, work in synergy to achieve robust, detail-aware detection on high-resolution images.

- **A New High-Resolution Benchmark:** We introduce the **HiRes-50K** dataset, featuring 50,568 images of up to 64 megapixels with paired sizes and JPEG compression levels, providing a more realistic and challenging benchmark for future research.

- **State-of-the-Art Performance:** HiDA-Net establishes a new state-of-the-art across multiple benchmarks, demonstrating significant gains of over 13% on the challenging Chameleon dataset and over 8% on our HiRes-50K, proving its superior robustness and generalization.

## 2 RELATED WORKS

### 2.1 EXISTING FEATURE EXTRACTION METHODOLOGIES

Existing detection methods differ in how they process input images to extract features related to synthetic content. Early approaches utilized traditional CNNs (Wang et al., 2020; Liu et al., 2020), while more recent works have leveraged pretrained vision-language or modern CNN models to capture global image inconsistencies (Liu et al., 2022). For example, UnivFD (Ojha et al., 2023) constructs a linear classifier using frozen CLIP features (Radford et al., 2021), while C2P-CLIP fine-tunes the model with carefully designed prompts (Tan et al., 2025). These resizing-based methods tend to suppress high-frequency components and degrade detection performance. To prevent this information loss, another line of research analyzes low-level features. Methods like PatchCraft (Zhong et al., 2023a) and AIDE (Yan et al., 2024a) propose strategies to select the most informative patches based on texture or frequency content. Similarly, TextureCrop (Konstantinidou et al., 2025) and SAFE (Li et al., 2024) demonstrated that cropping improves performance. However, these methods typically focus on limited regions, overlooking valuable information from the global context. B-Free processes multiple crops independently and obtains the final prediction by simply averaging their results (Guillaro et al., 2025). Our HiDA-Net addresses this by integrating local high-frequency details from all image tiles with global context in an end-to-end architecture.

### 2.2 RECONSTRUCTION-BASED DETECTION

Reconstruction error provides a strong detection signal. DIRE detects diffusion-generated images by first applying a diffusion-based noise and denoise reconstruction process, then computing the residual between the original and reconstructed image, which is used as input to a trainable classifier (Wang et al., 2023). Follow-up work simplifies this idea. Aeroblade finds that the pretrained autoencoder (AE) from a latent diffusion model alone reconstructs generated images with lower error than real ones, avoiding the costly denoising process (Ricker et al., 2024). DRCT operationalizes this by using diffusion-reconstructed real images as fake images in a contrastive framework, encouraging the model to learn subtle distinctions between real images and visually similar reconstructions (Chen et al., 2024). Rajan et al. (2025) used an autoencoder to construct paired training data. B-Free also uses self-conditioned reconstructions and content-aware inpainting of real images to construct bias-reduced paired training data (Guillaro et al., 2025). Inspired by this philosophy, we leverage VAE-based reconstructions and a randomized patch swap strategy to generate partially manipulated images with token-level labels, which we couple with our Token-wise Forgery Localization (TFL) objective to sharpen spatial sensitivity to localized edits.

### 2.3 GENERALIZATION ABILITY RESEARCH FOR DETECTION

Real world images undergo diverse degradations (e.g., JPEG recompression, resizing, blur), posing significant challenges to detector robustness. A common practice is to augment training data with different distortions (Wang et al., 2020; Yan et al., 2024a), yet this may teach the model tolerance to artifacts rather than distinguishing them from synthesis traces. Recent analyses reveal deeper biases that detectors often exploit dataset biases such as mismatched compression histories (Grommelt et al., 2025) or superficial distributional signals tied to source models or prompt styles (Zheng et al., 2024), hindering generalization beyond curated benchmarks. To mitigate these issues, we introduce a JPEG Quality Factor Estimation (QFE) task that encourages the model to separate compression noise from synthesis artifacts. We also propose **HiRes-50K**, a high-resolution dataset in which real and fake images are aligned in both image size and JPEG compression level, allowing for more realistic and controlled evaluation.

## 3 MOTIVATION: AN INFO-PRESERVING VIEW

Resizing introduces a strong low-pass effect that discards high-frequency cues critical for detection, while cropping preserves and redistributes this content via spectral leakage.

We compare two downsampling method on the SDv1.4 subset of GenImage: (i) resizing a $448 \times 448$ image to $224 \times 224$, and (ii) random cropping a $224 \times 224$ patch from the

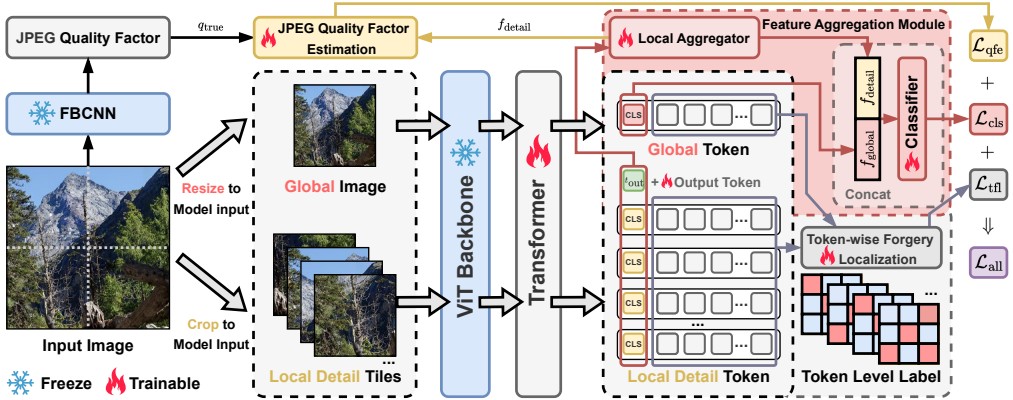

Figure 2: **Overview of HiDA-Net.** Input images are processed by two paths: (i) a *global path* that resizes $I$ to input size, and (ii) a *local path* that crops $K$ tiles. Both are fed into a *shared, frozen* ViT backbone and refined by a small trainable Transformer. The *[CLS]* tokens from all tiles are aggregated by the *Feature Aggregation Module (FAM)* for classification. Two tasks are trained jointly: *Token-wise Forgery Localization (TFL)* supervises patch tokens for localized manipulations, and *JPEG Quality Factor Estimation (QFE)* regresses the JPEG quality from $f_{\text{detail}}$.

original. For each method, we visualize the spectral energy ratio between real and generated images (Fig. 3). Red indicates higher energy for real images and blue indicates higher energy for generated ones. Real images exhibit stronger high-frequency components. Resizing truncates the outer regions of the frequency spectrum, removing high-frequency differences critical for detection. In contrast, random cropping retains high-frequency cues for detection.

Resizing an image $I$ is equivalent in the frequency domain to truncating its Discrete Fourier Transform (DFT), $\mathcal{F}\{I\}$, retaining only the central low-frequency coefficients, which irreversibly removes high-frequency information. When downsampling an image from $N_1 \times N_2$ to $M_1 \times M_2$ with an ideal low-pass filter, the DFT of the new image, $Y[r_1, r_2]$, is a centered crop of the original DFT $X[r_1, r_2]$, scaled by a constant:

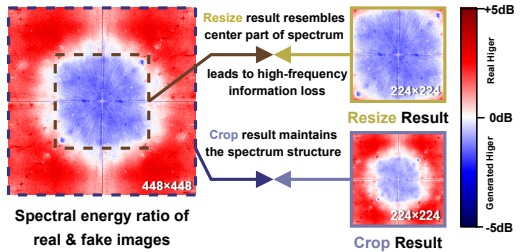

Figure 3: **Resizing vs Cropping in the Frequency Domain.** Visualization of spectral energy differences when downsampling from $448 \times 448$ to $224 \times 224$.

$$Y[r_1, r_2] = \left(\frac{M_1 \cdot M_2}{N_1 \cdot N_2}\right) X[r_1, r_2], |r_1| < \frac{M_1}{2}, |r_2| < \frac{M_2}{2}. \tag{1}$$

All high-frequency components beyond this central region are discarded.

In contrast, cropping a tile $P_k$ from $I$ is equivalent to multiplying $I$ by a window function $W_k$ of width $M_1$ and height $M_2$. By the convolution theorem:

$$D_M(\omega) = \frac{\sin(M\omega/2)}{\sin(\omega/2)}, \quad \mathcal{F}\{P_k\} = \mathcal{F}\{I \cdot W_k\} = \mathcal{F}\{I\} * \mathcal{F}\{W_k\}. \tag{2}$$

$$\mathcal{F}\{W_k\} = e^{-\frac{j\omega_1(M_1-1)}{2} + \frac{j\omega_2(M_2-1)}{2}} D_{M_1}(\omega_1) D_{M_2}(\omega_2). \tag{3}$$

where $*$ denotes convolution. The term $\mathcal{F}\{W_k\}$ is a Dirichlet kernel. This causes spectral leakage, effectively spreading information from all original frequencies, including high frequencies, across the entire spectrum of the cropped tile.

We further consider a partition of $I$ into $n_0 \times n_1$ *non-overlapping* tiles indexed by $(a, b)$, $a \in \{0, \ldots, n_0-1\}$ and $b \in \{0, \ldots, n_1-1\}$. The $(a, b)$-th tile has size $M_a^{(1)} \times M_b^{(2)}$ and top-left starting coordinates $\Delta_a^{(1)} = \sum_{i<a} M_i^{(1)}$ and $\Delta_b^{(2)} = \sum_{j<b} M_j^{(2)}$. Let $Y_{(a,b)}(e^{j\omega_1}, e^{j\omega_2})$ be the DTFT of tile

$(a, b)$. The DTFT of the full image can be reconstructed from the tiles via appropriate phase shifts:

$$X(e^{j\omega_1}, e^{j\omega_2}) = \sum_{a=0}^{n_0-1} \sum_{b=0}^{n_1-1} e^{-j(\omega_1 \Delta_a^{(1)} + \omega_2 \Delta_b^{(2)})} \cdot Y_{(a,b)}(e^{j\omega_1}, e^{j\omega_2}). \tag{4}$$

Thus, processing a sufficient set of tiles retains access to the **full frequency content** of the original image. Practically, we implement this by cropping tiles to cover the entire image, ensuring *full-spectrum coverage*. Compared with resizing, the cropped tiles collectively cover the full image and are fed into the model, preserving high-frequency information for accurate detection. The complete mathematical derivations can be found in the supplementary materials D.1.

## 4 METHODOLOGY

We propose the **Hi**gh-Resolution **D**etail-**A**ggregation Network (HiDA-Net), a dual-path detector for high-resolution images illustrated in Fig. 2. A *global view* provides semantic context, while a set of *lossless, full-resolution tiles* covers all pixels to preserve subtle high-frequency generation cues. Both paths share a frozen ViT backbone with a lightweight refinement layer. The *Feature Aggregation Module* (FAM) fuses tile-level and global *[CLS]* tokens for the final decision. To improve reliability in real-world conditions, we add two extra tasks: *Token-wise Forgery Localization* (TFL) for spatial sensitivity to localized edits, and *JPEG Quality Factor Estimation* (QFE) to disentangle compression artifacts from generative traces.

### 4.1 INPUT PREPROCESSING AND FEATURE EXTRACTION

Given an input image $I \in \mathbb{R}^{H \times W \times 3}$ of arbitrary resolution, we extract its features via two parallel paths:

**Global Path:** The image $I$ is resized to the standard input dimension of our backbone network (a Vision Transformer(ViT) with $224 \times 224$ input), yielding a global-view image $I_{\text{global}}$, which provides overall semantic information.

**Local Path:** To preserve high-frequency details, the original image $I$ is divided into $K$ tiles $\{I_1, I_2, \ldots, I_K\}$, each of size $224 \times 224$. These tiles are cropped directly from the source image without any resizing, thus maintaining pixel-level information. The tiling strategy ensures that the union of all tiles fully covers the original image.

Both the global image $I_{\text{global}}$ and the local tiles $\{I_k\}_{k=1}^K$ are passed through a *shared and frozen* pre-trained ViT backbone. The output tokens from the ViT's final layer are then refined by a small, trainable Transformer for subsequent tasks. For a tile $I_k$, this produces $T_k = \{t_{\text{cls}}^k, t_1^k, \ldots, t_N^k\}$, where $t_{\text{cls}}^k$ is the *[CLS]* token and the others are ViT image patch tokens. For the global image $I_{\text{global}}$, we obtain $T_{\text{global}} = \{t_{\text{cls}}^{\text{global}}, t_1^{\text{global}}, \ldots, t_N^{\text{global}}\}$. During training, we randomly sample $K \in [K_{\min}, K_{\max}]$ tiles to encourage robustness, and during inference, we deterministically cover the whole image to ensure no area is missed.

### 4.2 FEATURE AGGREGATION MODULE (FAM)

We fuse global semantics and high-fidelity local details to make the final classification.

**Local Detailed Feature Aggregation:** We collect the *[CLS]* tokens from all local patches to form a variable length sequence $\{t_{\text{cls}}^1, t_{\text{cls}}^2, \ldots, t_{\text{cls}}^K\}$. A lightweight Transformer encoder **Local Aggregator** encodings processes this sequence. We prepared a learnable output token $t_{\text{out}}$ and get the aggregated local detailed feature:

$$f_{\text{detail}} = \text{LocalAggregator}([t_{\text{out}}, t_{\text{cls}}^1, \ldots, t_{\text{cls}}^K])[0]. \tag{5}$$

**Global-Detail Fusion and Classification:** We concatenate the global *[CLS]* token $f_{\text{global}} = t_{\text{cls}}^{\text{global}}$ and $f_{\text{detail}}$ to obtain the final discriminative feature vector $f_{\text{final}}$:

$$f_{\text{final}} = \text{Concat}(f_{\text{global}}, f_{\text{detail}}) \tag{6}$$

which is fed to an MLP head to produce the binary probability $p$. $y_{\text{true}} \in \{0, 1\}$ indicates real/fake. The loss is:

$$\mathcal{L}_{\text{cls}} = \text{CrossEntropy}(p, y_{\text{true}}) \tag{7}$$

### 4.3 TOKEN-WISE FORGERY LOCALIZATION (TFL)

We introduce the Token-wise Forgery Localization (TFL) task to provide token-level supervision for localized manipulations. We adopt a **Random Patch Swap (RPS)** augmentation. For a given pair of real and fake images, we randomly swap a proportion of the corresponding image to form a composite with both real and fake regions. When image pairs are unavailable, we swap patches between a random real and a random fake image. This augmentation yields a "soft" label $y_{\text{token}} \in [0, 1]$ for each ViT patch token, computed as the average of binary pixel labels within the corresponding patch. See supplementary materials for more details.

For all non *[CLS]* tokens $t_i^k$ from both local tile $I_k$ and the global image $I_{\text{global}}$, a shared linear head with a Sigmoid function predicts the token's forgery probability $p_{\text{token},i}^k$. The TFL loss $\mathcal{L}_{\text{tfl}}$ is the mean Binary Cross-Entropy (BCE) over all tokens:

$$\mathcal{L}_{\text{tfl}} = \frac{1}{M_{\text{total}}} \sum_{k,i} \text{BCE}(p_{\text{token},i}^k, y_{\text{token},i}^k) \tag{8}$$

where $M_{\text{total}}$ is the total number of patch tokens.

### 4.4 JPEG QUALITY FACTOR ESTIMATION (QFE)

To improve the model's robustness against JPEG compression, we introduce the QFE task, which trains the model to actively perceive the degree of compression. Using the aggregated local feature $f_{\text{detail}}$, which is rich in high-frequency details most affected by compression. We regress the JPEG Quality Factor (QF) via:

$$q_{\text{pred}} = \text{MLP}_{\text{qf}}(f_{\text{detail}}). \tag{9}$$

Since some training images were compressed and then saved as a lossless format like PNG, we do not rely on file metadata. Instead, a pre-trained estimator in FBCNN (Jiang et al., 2021) to provides $q_{\text{true}}$ for supervision, and the loss is:

$$\mathcal{L}_{\text{qfe}} = \text{MSE}(q_{\text{pred}}, q_{\text{true}}). \tag{10}$$

This guides the model to distinguish grid-like quantization artifacts and disentangle content from compression during classification.

**Overall Loss** The final loss function is as below, $\alpha$ and $\beta$ are set to 1 in default.

$$\mathcal{L}_{\text{all}} = \mathcal{L}_{\text{cls}} + \alpha \mathcal{L}_{\text{tfl}} + \beta \mathcal{L}_{\text{qfe}}. \tag{11}$$

## 5 HIRES-50K: A NEW HIGH-RESOLUTION BENCHMARK FOR AIGI DETECTION

**Dataset Overview.**

To evaluate detectors under realistic high-resolution conditions, we introduce **HiRes-50K**, a challenging benchmark of high-quality images collected from accessible AIGI communities (Freepik, 2025; LiblibAI, 2025; Civitai, 2025) and a real-image community (Unsplash, 2025). The collection complied with the Terms of Service and privacy policies of each source at the time of access. Unlike datasets generated from a fixed model and a small set of prompts, HiRes-50K is built from in-the-wild AI image

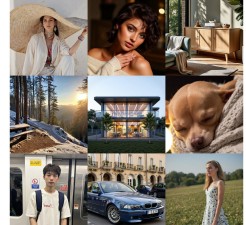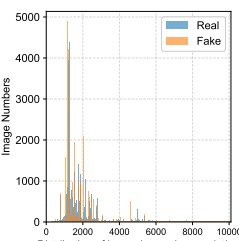

Figure 4: **Samples images and resolution distribution in HiRes-50K.** Left: generated images. Right: image long-edge resolution distribution.

Table 1: Cross-model accuracy (Acc) performance on the Chameleon testset. For each training dataset, the first row indicates the Acc evaluated on the Chameleon testset, and the second row gives the Acc for "fake image/real image" for detailed analysis.

| Training Dataset | CNNSpot | FreDect | UnivFD | DIRE | PatchCraft | NPR | AIDE | Ours |
|---|---|---|---|---|---|---|---|---|
| SD v1.4 | 60.11 | 56.86 | 55.62 | 59.71 | 56.32 | 58.13 | 62.60 | **73.44** |
| | 8.86/98.63 | 1.37/98.57 | 17.65/93.50 | 11.86/95.67 | 3.07/96.35 | 2.43/100.00 | 20.33/94.38 | 65.47/79.44 |
| All GenImage | 60.89 | 57.22 | 60.42 | 57.83 | 55.70 | 57.81 | 65.77 | **79.10** |
| | 9.86/99.25 | 0.89/99.55 | 85.52/41.56 | 2.09/99.73 | 1.39/96.52 | 1.68/100.00 | 26.80/95.06 | 76.77/82.20 |

sharing platforms and manually curated to include "human-hard", visually high-quality cases, making its distribution much closer to what detectors encounter on the real internet.

HiRes-50K includes 50,568 images spanning long-edge resolutions from below 1K to over 10K pixels, with some reaching **up to 64 megapixels**. For comparison, the recent Chameleon dataset comprises only 26K images, with most long edges resolution below 1,500 pixels (Yan et al., 2024a). Our benchmark surpasses it in both resolution and scale, significantly extending the evaluation boundary for high-resolution detection. As shown in Fig. 4, HiRes-50K features diverse content including portraits, landscapes, architecture, vehicles, and animals. We divide the dataset into eight resolution subsets: $[0, 900)$, $[900, 1200)$, $[1200, 1500)$, $[1500, 2000)$, $[2000, 2500)$, $[2500, 3000)$, $[3000, 5000)$, and $[5000, \infty)$, covering both common and extreme cases for stress-testing robustness.

**Construction principles.** We estimate the JPEG compression level of generated images and apply similar compression to low-compression real images after resizing them to match in pixel count. This process aligns real and fake images in both size and JPEG compression level within each resolution subset. Each subset maintains a 1:1 balance between real and fake images.

# 6 EXPERIMENT

## 6.1 IMPLEMENTATION DETAILS

**Input Tile Split** We use the pre-trained CLIP ViT-L/14 model as the backbone (Radford et al., 2021). We set the cropping resolution to $224 \times 224$ to align with the input resolution. Images with either side less than 224 are resized to 224 on the shorter side with preserved aspect ratio before applying tile cropping and augmentation. Other images keep their original resolution. During training, we randomly crop 1 to 16 tiles from one input image. During inference, we adopt a full-coverage tiling strategy. Specifically, if the input image has length $L$ along a spatial dimension and the tile size is $P$, we generate $N = \lceil \frac{L}{P} \rceil$ tiles along that dimension. The starting position of the $i$-th tile is:

$$x_i = \left\lfloor \frac{L}{N} \cdot (i-1) \right\rfloor, \quad \text{for } i = 1, 2, \ldots, N-1. \tag{12}$$

and the last tile starts at $x_N = L - P$. This ensures that the entire image is fully covered without missing any pixels.

**Data Augmentation** We adopt data augmentations including random JPEG compression (QF$\sim U(60, 100)$),random Gaussian Blur ($\sigma \sim U(0.1, 2.5)$), random scaling (scale factor$\sim U(0.25, 2)$) and Random Patch Swap augmentation (Swap ratio$\sim U(0.2, 0.98)$) to improve robustness. Each augmentation is conducted with 10% probability. Inspired by prior works (Rajan et al., 2025; Chen et al., 2024), we synthesize paired fake images on certain benchmarks using the VAE of a diffusion model that was used to generate the corresponding training data, which helps improve performance.

## 6.2 COMPARISON WITH STATE-OF-THE-ART METHODS

We conducted cross-model experiments on four different datasets to evaluate the generalizability of our method. Accuracy (ACC) is calculated using a threshold of 0.5.

**Comparisons on Chameleon** The Chameleon dataset poses a significant challenge for AIGI detection due to its high-resolution and visually indistinguishable synthetic images. As shown in Table 1, the previous SOTA method achieves only 65.77% accuracy (Yan et al., 2024a), with most existing methods achieving results below 60%, indicating limited discriminative capacity under such

Table 2: Cross-model accuracy (Acc) performance on our HiRes-50K Dataset.

| Resolution Range | 0-900 | 900-1200 | 1200-1500 | 1500-2000 | 2000-2500 | 2500-3000 | 3000-5000 | >5000 | Avg |
|---|---|---|---|---|---|---|---|---|---|
| CNNSpot Wang et al. (2020) | 58.46 | 56.37 | 58.73 | 63.14 | 67.42 | 60.90 | 66.30 | 58.90 | 61.28 |
| FreDect Frank et al. (2020) | 63.67 | 55.63 | 57.31 | 58.90 | 58.16 | 67.78 | 57.23 | 51.06 | 59.72 |
| UnivFD Ojha et al. (2023) | 67.00 | 58.35 | 62.20 | 65.95 | 66.05 | 58.20 | 58.15 | 60.05 | 62.05 |
| DIRE Wang et al. (2023) | 59.11 | 64.25 | 62.10 | 66.66 | 75.84 | 63.40 | 69.31 | 62.18 | 65.36 |
| AIDE Yan et al. (2024a) | 65.87 | 57.29 | 49.90 | 58.23 | 51.88 | 65.31 | 61.04 | 42.16 | 56.46 |
| DRCT Chen et al. (2024) | 65.30 | 66.19 | 68.78 | 68.85 | 68.78 | 75.79 | 69.65 | 54.03 | 67.17 |
| TextureCrop Konstantinidou et al. (2025) | 57.45 | 52.81 | 59.83 | 63.17 | 65.31 | 55.56 | 70.65 | 69.49 | 60.67 |
| RINE Koutlis & Papadopoulos (2024) | 65.21 | 60.30 | 63.60 | 63.02 | 58.60 | 61.30 | 61.87 | 63.77 | 62.21 |
| SPAI Karageorgiou et al. (2025) | 79.76 | 76.50 | 74.01 | 73.72 | 71.80 | 65.50 | 69.31 | 64.41 | 71.87 |
| Ours | **82.98** | **81.39** | **78.16** | **78.99** | **88.26** | **80.18** | **82.88** | **69.84** | **80.33** |

Table 3: Cross-model accuracy (Acc) performance on the GenImage Dataset.

| Method | Midjourney | SDv1.4 | SDv1.5 | ADM | GLIDE | Wukong | VQDM | BigGAN | Avg |
|---|---|---|---|---|---|---|---|---|---|
| Swin-T Liu et al. (2021) | 62.1 | 99.9 | 99.8 | 49.8 | 67.6 | 99.1 | 62.3 | 57.6 | 74.8 |
| CNNSpot Wang et al. (2020) | 52.8 | 96.3 | 95.9 | 50.1 | 39.8 | 78.6 | 53.4 | 46.8 | 64.2 |
| Spec Zhang et al. (2019) | 52.0 | 99.4 | 99.2 | 49.7 | 49.8 | 94.8 | 55.6 | 49.8 | 68.8 |
| F3Net Qian et al. (2020) | 50.1 | 99.9 | 99.9 | 49.9 | 50.0 | 99.9 | 49.9 | 49.9 | 68.7 |
| UnivFD Ojha et al. (2023) | 93.9 | 96.4 | 96.2 | 71.9 | 85.4 | 94.3 | 81.6 | 90.5 | 88.8 |
| NPR Tan et al. (2024b) | 81.0 | 98.2 | 97.9 | 76.9 | 89.8 | 96.9 | 84.1 | 84.2 | 88.6 |
| FreqNet Tan et al. (2024a) | 89.6 | 98.8 | 98.6 | 66.8 | 86.5 | 97.3 | 75.8 | 81.4 | 86.8 |
| FatFormer Liu et al. (2024) | 92.7 | 100.0 | 99.9 | 75.9 | 88.0 | 99.9 | 98.8 | 55.8 | 88.9 |
| DRCT Chen et al. (2024) | 91.5 | 95.0 | 94.4 | 79.4 | 89.1 | 94.6 | 90.0 | 81.6 | 89.4 |
| AIDE Yan et al. (2024a) | 79.4 | 99.7 | 99.8 | 78.5 | 91.8 | 98.7 | 80.3 | 66.9 | 86.8 |
| Effort Yan et al. (2024b) | 82.4 | 99.8 | 99.8 | 78.7 | 93.3 | 97.4 | 91.7 | 77.6 | 91.1 |
| SAFE Li et al. (2024) | 95.3 | 99.4 | 99.3 | 82.1 | 96.3 | 98.2 | 96.3 | 97.8 | 95.6 |
| C2P-CLIP Tan et al. (2025) | 88.2 | 90.9 | 97.9 | 96.4 | 99.0 | 98.8 | 96.5 | 98.7 | 95.8 |
| Ours/No VAE | 97.8 | 98.4 | 98.3 | 86.2 | 98.0 | 98.4 | 95.6 | 96.2 | 96.1 |
| Ours/SDv1.4 | 94.2 | 99.1 | 99.2 | 92.7 | 99.1 | 98.0 | 96.9 | 97.8 | **97.1** |

a high-resolution scenario. In contrast, our proposed HiDA-Net, specifically expert in handling high-resolution inputs, delivers substantial performance improvements. Notably, even when trained exclusively on the SD v1.4 subset, HiDA-Net outperforms competing methods trained on the entire GenImage dataset, while maintaining balanced accuracy across both real and generated samples. When trained on the full GenImage dataset, HiDA-Net achieves an accuracy of **79.10%**, outperforming the previous best by a substantial margin of **13%**.

**Comparisons on HiRes-50K.** To evaluate performance under challenging high-resolution conditions, we conduct experiments on our HiRes-50K dataset. All models are trained on the full GenImage training sets and evaluated on HiRes-50K. Results are summarized in Table 2. Our method consistently outperforms all competing approaches. Relative to the input-resizing baseline, it yields an average gain of **13.16%**. Compared with TextureCrop (Konstantinidou et al., 2025), which also crops inputs but selects only limited regions and averages the predictions of every tile, it shows improved performance at higher resolutions. The recent SPAI (Karageorgiou et al., 2025) method exhibits a gradual performance drop as dataset resolution increases, whereas our approach remains more stable and achieves over an **8%** performance gain compared to SPAI on average. Experiment results show our method's scalability and robustness.

**Comparisons on GenImage** Our method achieves consistently strong performance across both high-resolution and standard low-resolution benchmarks. Following the evaluation protocol of PatchCraft (Zhong et al., 2023a), all models are trained on the GenImage SD v1.4 subset and evaluated on the full GenImage dataset. As reported in Table 3, our approach yields results comparable to the current SOTA, C2P-CLIP (Tan et al., 2025), on individual subsets and surpasses it in terms of average accuracy. Moreover, incorporating fake-real image pairs synthesized by the SD v1.4 VAE further boosts performance by 1.3%, achieving a new SOTA.

**Comparisons on DRCT.** All models are trained on the DRCT SD v1.4 subset and evaluated on all testsets. As shown in Table. 4, DRCT achieves high accuracy by reconstructing each training image to form paired hard examples and applying contrastive training, which reveals subtle diffusion artifacts. Our plain HiDA-Net surpasses the DRCT with SD v1.4 reconstruction but falls short of the SD v2 variant. But by adopting a similar strategy, we construct paired training data with the SD v1.4 VAE, matching each real image to a subtly different synthetic counterpart, and apply Random Patch Swap (RPS) augmentation. The resulting model performs strongly, especially for high-resolution

Table 4: Cross-model accuracy performance on the testing subsets of DRCT. See supplementary materials for more comparison.

| Method | LDM | SDv1.4 | SDv1.5 | SDv2 | SDXL | SDXL Refiner | SD Turbo | SDXL Turbo | LCM SDv1.5 | LCM SDXL | SDv1 Ctrl | SDv2 Ctrl | SDXL Ctrl | SDv1 DR | SDv2 DR | SDXL DR | Avg |
|---|---|---|---|---|---|---|---|---|---|---|---|---|---|---|---|---|---|
| CNNSpot | 99.9 | 99.9 | 99.9 | 97.6 | 66.3 | 86.6 | 86.2 | 72.4 | 98.3 | 61.7 | 98.0 | 85.9 | 82.8 | 60.9 | 51.4 | 50.3 | 81.1 |
| F3Net | 99.9 | 99.8 | 99.8 | 88.7 | 55.9 | 87.4 | 68.3 | 63.7 | 97.4 | 55.0 | 98.0 | 72.4 | 82.0 | 65.4 | 50.4 | 50.3 | 77.1 |
| Conv-B | 99.9 | 100.0 | 99.9 | 95.8 | 64.4 | 82.0 | 80.8 | 60.8 | 99.2 | 62.3 | 99.8 | 83.4 | 73.3 | 61.7 | 51.8 | 50.4 | 79.1 |
| UnivFD | 98.3 | 96.2 | 96.3 | 93.8 | 91.0 | 93.9 | 86.4 | 85.9 | 90.4 | 89.0 | 90.4 | 81.1 | 89.1 | 52.0 | 51.0 | 50.5 | 83.5 |
| DRCT/SDv1.4 | 99.9 | 99.9 | 99.9 | 96.3 | 83.9 | 85.6 | 91.9 | 70.0 | 99.7 | 78.8 | 99.9 | 95.0 | 81.2 | 99.9 | 95.4 | 75.4 | 90.8 |
| DRCT/SDv2 | 99.7 | 98.6 | 98.5 | 99.9 | 96.1 | 98.7 | 99.6 | 83.3 | 98.5 | 93.8 | 96.7 | 99.9 | 97.7 | 93.9 | 99.9 | 90.4 | 96.6 |
| Ours/No VAE | 98.7 | 98.8 | 98.8 | 98.7 | 98.8 | 98.8 | 97.7 | 97.6 | 98.6 | 98.8 | 98.8 | 98.4 | 98.2 | 90.4 | 74.8 | 71.2 | 94.8 |
| Ours/SDv1.4 | 98.8 | 98.9 | 98.9 | 98.0 | 99.0 | 98.8 | 98.5 | 98.8 | 98.4 | 98.9 | 98.5 | 98.7 | 98.4 | 99.0 | 97.4 | 94.9 | **98.4** |

Table 5: Cross-model AUC performance on mixed test set.

| Image Size | < 0.5 MPixels | | | 0.5 - 1.0 MPixels | | | | | | > 1.0 MPixels | | | | AVG |
|---|---|---|---|---|---|---|---|---|---|---|---|---|---|---|
| Approach | Glide | SD1.3 | SD1.4 | Flux | DALLE2 | SD2 | SDXL | SD3 | GigaGAN | MJv5 | MJv6.1 | DALLE3 | Firefly | |
| Fusing | 63.0 | 62.8 | 62.2 | 57.5 | 76.7 | 66.9 | 62.1 | 38.8 | 80.4 | 64.0 | 74.0 | 25.2 | 76.3 | 62.3 |
| LGrad | 76.5 | 82.4 | 83.4 | 74.9 | 85.7 | 60.7 | 70.2 | 12.7 | 89.9 | 69.2 | 79.6 | 30.0 | 42.0 | 65.9 |
| UnivFD | 63.3 | 80.8 | 81.2 | 36.3 | 91.4 | 84.3 | 78.3 | 28.6 | 86.2 | 57.1 | 60.5 | 31.0 | 95.5 | 67.3 |
| GramNet | 78.2 | 83.9 | 84.3 | 78.6 | 85.2 | 66.7 | 77.8 | 19.2 | 85.0 | 63.8 | 84.9 | 42.9 | 38.0 | 68.4 |
| DeFake | 86.1 | 64.2 | 63.6 | 90.5 | 41.4 | 66.2 | 52.3 | 87.7 | 71.7 | 67.0 | 87.5 | 93.3 | 39.4 | 70.1 |
| PatchCraft | 78.4 | 95.7 | 96.2 | 86.9 | 81.8 | 95.7 | 96.7 | 33.8 | 98.0 | 79.0 | 96.1 | 28.1 | 79.1 | 80.4 |
| DMID | 73.1 | 100.0 | 100.0 | 97.2 | 54.3 | 99.7 | 99.6 | 67.9 | 67.9 | 99.9 | 94.4 | 41.3 | 90.2 | 83.5 |
| RINE | 95.6 | 99.9 | 99.9 | 93.0 | 93.0 | 96.6 | 99.3 | 39.1 | 92.9 | 96.4 | 81.2 | 41.8 | 82.9 | 85.5 |
| SPAI | 90.2 | 99.6 | 99.6 | 83.0 | 91.1 | 96.5 | 97.4 | 75.9 | 85.4 | 94.5 | 84.0 | 90.2 | 96.0 | **91.0** |
| Ours | 99.4 | 99.6 | 99.6 | 88.1 | 91.2 | 96.8 | 98.5 | 45.6 | 95.1 | 92.6 | 90.6 | 80.6 | 89.4 | 89.8 |

SDXL images at $1024 \times 1024$ and DR variants (partially inpainted images), achieving a state-of-the-art average accuracy of 98.4%.

**Comparisons on Mix Set.** Following SPAI (Karageorgiou et al., 2025), we train our model on a single latent diffusion model (Rombach et al., 2022) and evaluate AUC on the same Mixed Set. Our method consistently outperforms other baselines and is only 0.2% below SPAI. Although the maximum resolution in Mix Set is just above 1 megapixel, on the higher-resolution Hires-50K dataset our approach exceeds SPAI by over 8% in ACC, underscoring its advantage on high-resolution images.

## 6.3 ROBUSTNESS & ABLATION STUDY

Table 6: Ablation study on the number of crop tiles in FAM.

| Tile Nums | 1 | 2 | 4 | 8 | 16 | FAM |
|---|---|---|---|---|---|---|
| ACC (%) | 92.14 | 93.34 | 95.63 | 95.69 | 95.89 | **96.10** |

Table 7: Ablation study of TFL and QFE tasks on GenImage.

| Module | FAM | FAM+TFL | FAM+QFE | ALL |
|---|---|---|---|---|
| ACC (%) | 93.92 | 94.36 | 94.73 | **96.10** |

**Robustness Evaluation** We conducted a series of experiments to assess the robustness of our method against common image perturbations. We trained our model on the SD v1.4 subset of GenImage and evaluated it under various perturbations, including JPEG compression, Gaussian blur, and image scaling, and the results are summarized in Fig. 5. We also examine the effect of Gaussian noise perturbations, and report the results in Table 8.

For JPEG compression, we evaluate robustness in two settings: testing on the SD v1.4 subset and on the entire GenImage test set. For Gaussian blur and scaling perturbations, evaluations are performed on the SD v1.4 subset only. This broader evaluation shows that our method generalizes well across varying compression levels and datasets. In all cases, our method demonstrates strong resilience, maintaining high accuracy under moderate degradation. These results validate the robustness and generalization capabilities of our approach in real-world conditions.

**Effect of FAM Module** We conduct an ablation study on the number of cropped tiles sent to the FAM module during both training and inference. Evaluation uses the same settings as comparisons on the GenImage. We evaluate fixed sampling strategies with $n \in \{1, 2, 4, 8, 16\}$ tiles. When $n = 1$, a single center crop is used when inference. For other cases, tiles are randomly sampled during both training and inference. In contrast, our FAM Module samples 1–16 tiles during training

Table 8: Model accuracy against Gaussian noise perturbation on Genimage dataset.

| Noise Std | 0 | 0.002 | 0.004 | 0.006 | 0.008 | 0.01 |
|---|---|---|---|---|---|---|
| ACC (%) | 96.10 | 94.95 | 92.65 | 89.92 | 0.8877 | 88.23 |

Table 9: Ablation study of Global and Local branches.

| Module | No Global | No Local | ALL |
|---|---|---|---|
| ACC (%) | 94.75 | 91.88 | **96.10** |

Figure 5: Model Accuracy Against Diverse Perturbation.

and uses a full-coverage crop during inference. As shown in Table. 6, increasing the number of tiles leads to better performance. FAM's full-coverage strategy achieves the best performance, validating that aggregating more high-frequency tiles benefits detection. In the FAM module, both the local and global tiles play crucial roles, and their combination yields the best overall performance on the GenImage dataset, as shown in Table 9. The global tiles provide a holistic, image-level context that cannot be captured by any single local region, whereas the local tiles are better suited for modeling fine-grained spatial details and therefore perform particularly well at higher input resolutions.

**Effectiveness of TFL and QFE** We ablate the effects of Token-wise Forgery Localization (TFL) and JPEG Quality Factor Estimation (QFE) using the same GenImage setup. As shown in Table 7, both modules independently improve performance, their combination achieves the best results. QFE's impact on JPEG robustness is visualized in Fig. 5. QFE shows limited gains on SD v1.4 since the model was trained and tested on the same JPEG augmented SD v1.4 data. But it greatly improves robustness on the unseen GenImage set, showing better generalization. Table 4 shows that combining TFL with Random Patch Swap (RPS) further enhances its sensitivity to partially inpainted fine-grained manipulations, yielding strong performance.

### 6.4 COMPUTATIONAL COMPLEXITY

Our proposed HiDA-Net contains 331.05M parameters, representing only a modest increase over the 303.97M parameters of the ViT backbone. For an input image of resolution $224 \times 224$, the computational cost is 343.3 GFLOPs, which is approximately twice that of the corresponding ViT backbone (162.2 GFLOPs), since our architecture extracts both global tiles and local tiles at this resolution. For higher-resolution inputs, the computational cost grows approximately linearly with the number of tiles, as illustrated in Fig. 6.

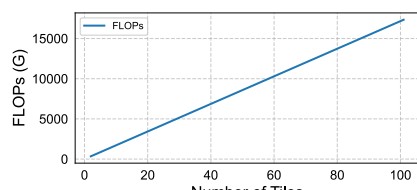

Figure 6: FLOPs vs. Number of Tiles

### 7 CONCLUSION & LIMITATION

In this paper, we propose *HiDA-Net*, a network designed to detect high-resolution AI-generated images without sacrificing fine-grained artifacts. It fuses features from full-resolution local tiles and a global context view via the Feature Aggregation Module (FAM), and introduces two additional tasks: Token-wise Forgery Localization (TFL) and JPEG Quality Factor Estimation (QFE). Our model achieves state-of-the-art performance across multiple benchmarks, with notable gains on the challenging Chameleon dataset. To support future research, we introduce *HiRes-50K*, a high-resolution benchmark. While HiDA-Net demonstrates strong effectiveness, a key limitation is the increased inference time on large images due to tile-wise processing. Our future work will optimize tile fusion and efficiency for real-world deployment.

ACKNOWLEDGEMENTS

This work is supported by the National Natural Science Foundation of China (Grant No. U21B2004)

REPRODUCIBILITY STATEMENT

We have taken extensive measures to facilitate the reproducibility of our work. The main text and appendix provide detailed descriptions of the algorithms, optimization procedures, ablation studies, hyperparameter settings, and experimental protocols to ensure transparency.

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

## A    APPENDIX

Our implementation is included in the *code* folder of the supplementary zip. The HiRes-50K dataset will be released soon, and we plan to open-source the full codebase.

## B    IMPLEMENTATION DETAILS

**Training Platform.** All experiments are conducted on a server equipped with dual AMD EPYC 7543 CPUs, 512 GB of RAM, and four NVIDIA RTX 3090 GPUs.

**Model Parameters.** We utilize the ViT-L/14 variant of the pretrained vision backbone from OpenAI CLIP Radford et al. (2021) as our feature extractor. Our model incorporates a lightweight, trainable feature refinement Transformer composed of 2 layers and 2 attention heads. The local aggregation modules are implemented with a single Transformer layer and 2 attention heads. Additionally, we employ a 2-layer token-level classification MLP and a 2-layer output classification MLP with ReLU activation functions. For JPEG quality factor estimation, we adopt a 4-layer MLP.

**Training Settings.** The model is optimized using the AdamW optimizer with a base learning rate of 1e-4 and a weight decay of 0.05. We use a dropout rate of 0.1 throughout the network to prevent overfitting. The batch size is set to 64, and the model is trained for a total of 10 epochs. For the loss function, we set $\alpha = \beta = 1$.

**Augmentation Settings.** We adopt data augmentations including random JPEG compression (QF$\sim U(60, 100)$),random Gaussian Blur ($\sigma \sim U(0.1, 2.5)$), random scaling (scale factor$\sim U(0.25, 2)$) and Random Patch Swap augmentation (Swap ratio$\sim U(0.2, 0.98)$) to improve robustness. Each augmentation is conducted with 10% probability. Inspired by prior works Rajan et al. (2025); Chen et al. (2024), we further synthesize paired fake images on certain benchmarks using the VAE component of the diffusion model originally used to generate the data. Specifically, we use the VAE from Stable Diffusion v1.4 to perform these augmentations.

## C    INTRODUCTION FOR THE HIRES-50K DATASET

Table 10: **Comparisons between different testsets.** Our `HiRes-50K` dataset encompasses high-quality, real-world scenarios with a broad range of high resolutions, covering image dimensions from 512 pixels up to 10K in edge length. Compared to the AIGCDetectBenchmark Zhong et al. (2023b) and the latest Chameleon Yan et al. (2024a) dataset, ours surpasses them in resolution. (IN represents ImageNet.)

| | ProGAN | StyleGAN | BigGAN | CycleGAN | StarGAN | GauGAN | StyleGAN2 | WFIR | ADM | Glide | Midjourney | SD v1.4 | SD v1.5 | VQDM | Wukong | DALLE2 | Chameleon | HiRes-50K |
|---|---|---|---|---|---|---|---|---|---|---|---|---|---|---|---|---|---|---|
| Magnitude | 8.0k | 12.0k | 4.0k | 2.6k | 4.0k | 10.0k | 15.9k | 2.0k | 12.0k | 12.0k | 12.0k | 12.0k | 16.0k | 12.0k | 12.0k | 2.0k | 26.0k | **50.05k** |
| Resolution | 256 | 256 | 256 | 256 | 256 | 256 | 256 | 1024 | 256 | 256 | 1024 | 512 | 512 | 256 | 512 | 256 | 720P-4K | **500-10,000** |
| Variety | LSUN | LUSN | IN | IN | CelebA | COCO | LSUN | FFHQ | IN | IN | IN | IN | IN | IN | IN | IN | Real-life | **Category-Rich Real-life** |

### C.1    INTRODUCTION

With the rapid advancement of image generation models, the visual fidelity of AI-generated content has significantly improved. Image resolutions have progressed from $512 \times 512$ to over $1000 \times 1000$. Earlier methods were mostly trained and evaluated on low-resolution AI-generated images because image synthesis models were not able to produce high-resolution outputs at that time. To overcome the common limitations of existing AI-generated image detection datasets, such as low-resolution datasets and the simplistic nature of images generated from basic prompts, we introduce a high-quality dataset comprising high-resolution and diverse images that are often indistinguishable from real photographs to the

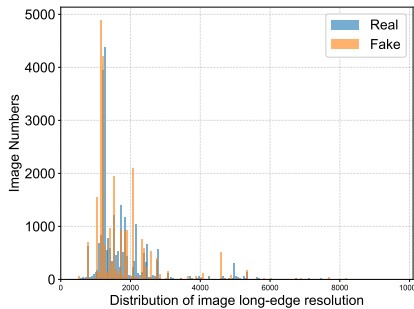

Figure 7: **Resolution distribution.** Image long-edge resolution distribution in our HiRes-50K dataset.

human eye, designed to evaluate detection models under high-resolution settings and support the development of more robust detection techniques.

## C.2 DATASET COMPOSITION

### C.2.1 AI-GENERATED IMAGES

The AI-Generated part contains over 25,000 images spanning diverse categories, including portraits, landscapes, architecture, vehicles and animals. Among these, portraits constitute the majority and encompass a wide range of subjects such as facial close-ups, upper-body shots, full-body images, group portraits, and human figures embedded in natural or social contexts. Landscape images cover environments such as mountains, beaches, cities, rural areas, and deserts under various weather conditions (e.g., clear skies, snow, sunrise, sunset). Architectural images mainly depict modern urban scenes, such as skyscrapers, villas, and urban neighborhoods. The sample images are shown in Figure 8.

The long side resolution of images ranges from under 900 pixels to over 10,000 pixels. To analyze resolution-specific performance, we group images into eight resolution bins: $[0, 900]$, $[900, 1200]$, $[1200, 1500]$, $[1500, 2000]$, $[2000, 3000]$, $[3000, 5000]$, $[5000, \infty]$, with corresponding sample counts of 845, 6665, 6399, 5262, 3674, 571, 1196 and 472, respectively. Notably, the majority of AI-Generated images fall within the $[900, 2500]$ range, which reflects typical image resolutions encountered online and exceeds those used in traditional benchmarks (e.g., 224–512 pixels). This broad resolution coverage supports comprehensive evaluation across scales. To ensure visual fidelity, we estimate JPEG quality factors for all images and filter out those below 75. This selection process removes low-quality content and guarantees a high-quality dataset suitable for rigorous detection model assessment.

### C.2.2 REAL IMAGES

To ensure a fair and balanced comparison, we align the real images with the generated ones in both resolution and JPEG compression level. For each image class, we first select high-quality real images with larger dimensions and higher fidelity than the corresponding AI-generated images. These real images are then resized to the total pixel count which matches the AI-generated images by multiplying a pixel scaling factor. This operation maintains the aspect ratio of real images. After resizing, we apply JPEG compression using the same quality levels observed in the synthetic data. This procedure ensures that within each subcategory, the real and AI-generated images have similar distributions in terms of both resolution and compression. We also maintain a one-to-one ratio between real and generated images for every class. As a result, our benchmark supports controlled and meaningful evaluations across the two domains. Sample images are shown in Figure 8.

## C.3 COMPARISONS WITH EXISTING DATASET

In Table 10, we present a comprehensive comparative analysis between our HiRes-50K dataset and AIGC test sets. The HiRes-50K dataset exhibits three distinguishing characteristics: (i) Magnitude. Encompassing 50,658 test images, our dataset constitutes the most extensive collection to date, significantly surpassing existing single test sets in scale and thereby enhancing robustness in evaluation. (ii) Variety. The data set incorporates a diverse array of real-world images, spanning categories such as portraits, landscapes, architecture, vehicles and animals. This breadth of coverage exceeds the narrow categorical scope of comparable datasets. (iii) Resolution. The images exhibit a wide resolution spectrum, ranging from 512 pixels to over 10K pixels on the long side. In summary, the HiRes-50K dataset establishes a more rigorous and pragmatically relevant benchmark, advancing the development of AI-generated image detection methodologies.

## C.4 DATA SOURCES

Our dataset is primarily constructed from several widely used online communities that provide either AI-generated or real images. We are grateful for their free and open contributions, which significantly support academic research in this field.

- **AI-generated Image Communities:**

- Freepik
- Liblib AI
- Civitai
- **Real Image Community:**
  - Unsplash

## C.5 LICENSING AND COMPLIANCE

We adhere strictly to the terms of service and licensing agreements associated with each image source:

- **Freepik:** All images were obtained through paid downloads under Freepik's licensing terms, which permit legal use of selected AI-generated content for research purposes.
- **Civitai:** We comply with the Civitai Terms of Service (Section 9.2), ensuring that all selected AI-generated images are publicly available and approved for reuse.
- **Liblib AI:** We follow the Liblib AI User Agreement (Section 6.2.2) and only use images that are openly accessible and permitted for public use.
- **Unsplash:** All real-world photographs from Unsplash are used under their openly licensed terms, which support research and educational purposes.

## D THEORETICAL PROOF OF INFO-PRESERVING

### D.1 FOURIER PRELIMINARIES

**DTFT/DFT.** For a 1D sequence $x[n]$, its DTFT and $N$-point DFT are

$$X(e^{j\omega}) = \sum_{n=-\infty}^{\infty} x[n]e^{-j\omega n}, \tag{13}$$

$$X[k] = \sum_{n=0}^{N-1} x[n]e^{-j\frac{2\pi}{N}kn}, \quad k = 0, \ldots, N-1. \tag{14}$$

The DFT samples the $2\pi$-periodic DTFT:

$$X[k] = X\left(e^{j\omega_k}\right), \quad \omega_k = \frac{2\pi k}{N}. \tag{15}$$

**Rectangular Window and Dirichlet Kernel.** Let the length-$M$ rectangular window be:

$$w_M[n] = \begin{cases} 1, & 0 \leq n \leq M-1, \\ 0, & \text{otherwise.} \end{cases} \tag{16}$$

Its DTFT is the Dirichlet kernel with a centering phase

$$W_M(e^{j\omega}) = e^{-j\omega\frac{M-1}{2}} D_M(\omega), D_M(\omega) = \frac{\sin\left(\frac{M\omega}{2}\right)}{\sin\left(\frac{\omega}{2}\right)}. \tag{17}$$

**DTFT under rate change (real $D > 0$).** We define decimation by a real factor $D > 0$ as bandlimited resampling: let $X_{\text{lp}}(e^{j\omega})$ be $X(e^{j\omega})$ low-pass filtered to $|\omega| < \pi/D$. Then

$$Y(e^{j\omega}) = \frac{1}{D} X_{\text{lp}}\left(e^{j\omega/D}\right), \quad |\omega| < \pi. \tag{18}$$

In the rational case $D = N/M$ with $M, N \in \mathbb{Z}_+$,

$$Y(e^{j\omega}) = \frac{M}{N} X_{\text{lp}}\left(e^{j\omega M/N}\right). \tag{19}$$

(Proof sketch: interpret $y[m]$ as ideal bandlimited interpolation of $x[n]$ followed by sampling on the grid $mD$; time-rate change by $D$ maps $\omega \mapsto \omega/D$ and introduces the factor $1/D$.)

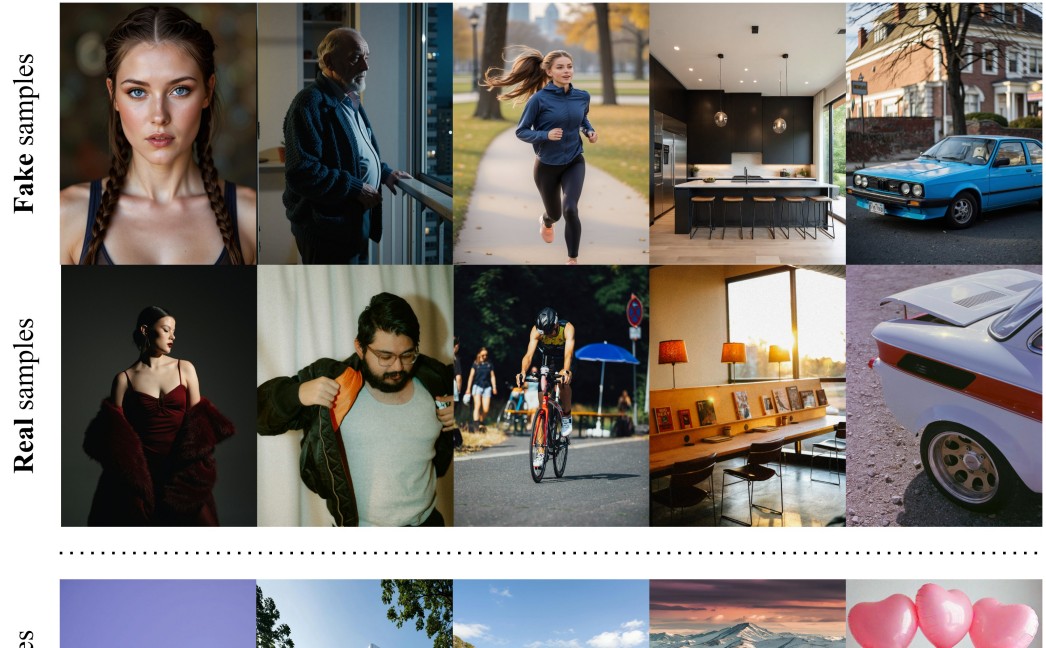

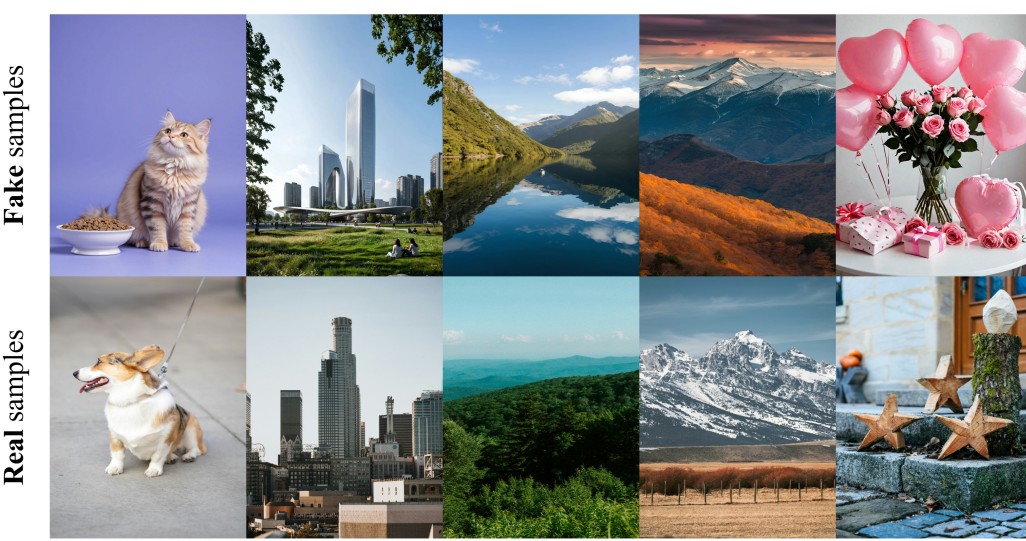

**Category-Rich Samples in HiRes-50K Dataset**

Figure 8: **HiRes-50K Dataset Samples.** Our HiRes-50K dataset combines various categories from real world scenarios, including human, architectures, animals, vehicles and landscapes. The first row displays the fake image collected from internet communities, while the second row indicates the real images.

## D.2 RESIZING AS LOW-PASS TRUNCATION

**1D derivation.** Evaluating the DTFT relationship from equation 19 on the respective DFT grids shows that the DFT of the resized signal is a scaled version of the low-frequency coefficients of the original signal's DFT. To formalize this, we consider the frequency indices centered around the DC component ($r = 0$). The relationship between the $M$-point DFT $Y[r]$ and the $N$-point DFT $X[r]$ is:

$$Y[r] = \frac{M}{N} X[r], \quad \text{for} \ -\frac{M}{2} \le r < \frac{M}{2}, \tag{20}$$

and is zero for all out-of-band frequencies. This explicitly states that the $M$-point spectrum is formed by taking the $M$ lowest-frequency components of the $N$-point spectrum (properly centered), scaling them, and discarding all others. This corrected formulation aligns with the standard understanding of ideal resampling and is consistent with the 2D case below.

**2D extension.** For an $N_1 \times N_2$ image downsampled by $(D_1, D_2)$ to $M_1 \times M_2$ $(D_i = \frac{N_i}{M_i})$ with ideal separable low-pass prefilters, the 2D DTFT obeys the rate-change rule, including the amplitude scaling:

$$Y(e^{j\omega_1}, e^{j\omega_2}) = \frac{M_1 M_2}{N_1 N_2} X_{\text{lp}}\left(e^{j\frac{\omega_1}{D_1}}, e^{j\frac{\omega_2}{D_2}}\right), \quad |\omega_i| < \pi. \tag{21}$$

Here, $X_{\text{lp}}$ is the low-pass filtered version of the original signal's DTFT. Sampling this relationship on the $M_1 \times M_2$ DFT grid and applying equation 15 along both axes gives the centered low-pass truncation:

$$Y[r_1, r_2] = \left(\frac{M_1 M_2}{N_1 N_2}\right) X[r_1, r_2], |r_1| < \tfrac{M_1}{2}, \ |r_2| < \tfrac{M_2}{2}, \tag{22}$$

and $Y[r_1, r_2] = 0$ outside this region. Thus, resizing preserves only the in-band (low-frequency) coefficients and irretrievably discards high-frequency content.

### D.3 CROPPING AS WINDOWING & SPECTRAL LEAKAGE

**1D crop.** Cropping the length-$M$ segment that starts at index $\Delta$ is the same as multiplying $x[n]$ by a *shifted rectangular window*:

$$v[n] = w_M[n - \Delta] \quad (\text{so } v[n] = 1 \text{ for } n \in [\Delta, \Delta + M - 1]),$$

i.e.,

$$y[n] = x[n] \, v[n]. \tag{23}$$

The DTFT of the shifted window is a linear phase times the Dirichlet kernel,

$$V(e^{j\omega}) = e^{-j\omega\Delta} W_M(e^{j\omega}) = e^{-j\omega\Delta} e^{-j\omega\frac{M-1}{2}} D_M(\omega), \tag{24}$$

where $W_M$ and $D_M$ are given in equation 17. By the convolution theorem:

$$Y(e^{j\omega}) = \frac{1}{2\pi} \int_{-\pi}^{\pi} X(e^{j\nu}) V\left(e^{j(\omega-\nu)}\right) d\nu = (X * V)(\omega) = \left(X * \left[e^{-j(\cdot)\Delta} W_M\right]\right)(\omega). \tag{25}$$

Since $W_M$ is the Dirichlet kernel, equation 25 convolves $X$ with a broad mainlobe plus sidelobes; this **spreads (leaks) energy from all frequencies** of $x$ across the spectrum of $y$. The linear phase $e^{-j\omega\Delta}$ only shifts phase and does not change magnitudes, so cropping preserves rather than removes high-frequency content. Hence the spectral-leakage effect that preserves high-frequency cues.

**2D crop.** For a tile of width $M_1$ and height $M_2$ with top-left offset $(\Delta_1, \Delta_2)$,

$$y[m_1, m_2] = I[\Delta_1 + m_1, \ \Delta_2 + m_2] \, w_{M_1}[m_1] \, w_{M_2}[m_2]. \tag{26}$$

Let

$$W_{M_1, M_2}(e^{j\omega_1}, e^{j\omega_2}) = e^{-j\omega_1 \frac{M_1-1}{2} - j\omega_2 \frac{M_2-1}{2}} D_{M_1}(\omega_1) D_{M_2}(\omega_2) \tag{27}$$

be the 2D window DTFT. Then

$$\mathcal{F}\{y\}(e^{j\omega_1}, e^{j\omega_2}) = \left(X(e^{j\omega_1}, e^{j\omega_2}) \, e^{j(\omega_1\Delta_1 + \omega_2\Delta_2)}\right) * W_{M_1, M_2}(e^{j\omega_1}, e^{j\omega_2}). \tag{28}$$

i.e., windowing induces Dirichlet-kernel convolution (spectral leakage) together with a phase term from the spatial offset.

### D.4 RECONSTRUCTION FROM CROPPED TILES

Consider a partition of $I$ into $n_0 \times n_1$ *non-overlapping* tiles indexed by $(a, b)$, with sizes $M_a^{(1)} \times M_b^{(2)}$ and top-left coordinates $\Delta_a^{(1)} = \sum_{i<a} M_i^{(1)}$ and $\Delta_b^{(2)} = \sum_{j<b} M_j^{(2)}$. Define the tile $(a, b)$ in its *local* coordinates by

$$y_{(a,b)}[m_1, m_2] = I\left(\Delta_a^{(1)} + m_1, \ \Delta_b^{(2)} + m_2\right), \quad 0 \le m_1 < M_a^{(1)}, \ 0 \le m_2 < M_b^{(2)}. \tag{29}$$

Its DTFT is

$$Y_{(a,b)}(e^{j\omega_1}, e^{j\omega_2}) = \sum_{m_1=0}^{M_a^{(1)}-1} \sum_{m_2=0}^{M_b^{(2)}-1} y_{(a,b)}[m_1, m_2]\, e^{-j(\omega_1 m_1 + \omega_2 m_2)}. \tag{30}$$

The DTFT of the full image equals the phase-aligned sum of tile DTFTs:

$$X(e^{j\omega_1}, e^{j\omega_2}) = \sum_{a=0}^{n_0-1} \sum_{b=0}^{n_1-1} e^{-j\left(\omega_1 \Delta_a^{(1)} + \omega_2 \Delta_b^{(2)}\right)} Y_{(a,b)}(e^{j\omega_1}, e^{j\omega_2}). \tag{31}$$

*Proof.* Embed each tile back to global coordinates via

$$x_{(a,b)}[n_1, n_2] \triangleq y_{(a,b)}\left(n_1 - \Delta_a^{(1)},\, n_2 - \Delta_b^{(2)}\right),$$

which equals $I[n_1, n_2]$ on the tile support and 0 elsewhere. Since the tiles are non-overlapping and cover the image,

$$I[n_1, n_2] = \sum_{a=0}^{n_0-1} \sum_{b=0}^{n_1-1} x_{(a,b)}[n_1, n_2].$$

(Linearity) Taking the DTFT yields

$$X(e^{j\omega_1}, e^{j\omega_2}) = \sum_{a=0}^{n_0-1} \sum_{b=0}^{n_1-1} \mathcal{F}\{x_{(a,b)}\}(e^{j\omega_1}, e^{j\omega_2}).$$

(Shift) By the 2D shift property,

$$\mathcal{F}\{x_{(a,b)}\}(e^{j\omega_1}, e^{j\omega_2}) = e^{-j(\omega_1 \Delta_a^{(1)} + \omega_2 \Delta_b^{(2)})} \cdot \mathcal{F}\{y_{(a,b)}\}(e^{j\omega_1}, e^{j\omega_2}). \tag{32}$$

Using equation 30, we have $\mathcal{F}\{y_{(a,b)}\} = Y_{(a,b)}$, hence

$$X(e^{j\omega_1}, e^{j\omega_2}) = \sum_{a=0}^{n_0-1} \sum_{b=0}^{n_1-1} e^{-j(\omega_1 \Delta_a^{(1)} + \omega_2 \Delta_b^{(2)})} Y_{(a,b)}(e^{j\omega_1}, e^{j\omega_2}), \tag{33}$$

which proves equation 31.

### D.5    PUTTING IT TOGETHER: CROP VS. RESIZE

Resizing (§D.2) truncates the spectrum per equation 22 and discards high-frequency content. Cropping (§D.3) multiplies the signal by a rectangular window whose DTFT is the Dirichlet kernel equation 17, yielding the convolution equation 25 that *redistributes* (rather than removes) high-frequency energy across the band. Finally, combining non-overlapping tiles with the phase factors in equation 31 exactly reconstructs the global DTFT, guaranteeing **full-spectrum coverage**.

Therefore, by operating on tiled crops with explicit full-spectrum coverage and phase-consistent aggregation, **HiDA-Net** is intrinsically suited to retain fine-grained high-frequency cues. While this analysis is based on ideal filters, its conclusion holds in practice, as standard resizing algorithms (e.g., bicubic interpolation) approximate this low-pass filtering and thus inevitably attenuate the very high-frequency details that cropping preserves via spectral leakage. This ability to exploit information across the full spectrum leads to stronger detection on high-resolution images.

## E    MORE COMPARISONS

To further assess the generalization ability of our method, we conduct additional experiments across four representative datasets. Accuracy (ACC) is reported using a fixed decision threshold of 0.5.

**Evaluation on Chameleon.** We first evaluate models trained on the Stable Diffusion v1.4 (SD v1.4) subset and the complete GenImage dataset. Testing is conducted on the Chameleon dataset, with

Table 11: **Dataset Details in Benchmark GenImage.**

|  | Dataset | Image Size | Number | Source |
|---|---|---|---|---|
| **Train** | SD v1.4 (StableDiffusion, 2022) | $512 \times 512$ | 324.0k | ImageNet Deng et al. (2009) |
| **Test** | BigGAN (Brock et al., 2019) | $256 \times 256$ | 12.0k | ImageNet (Deng et al., 2009) |
|  | ADM (Dhariwal & Nichol, 2021) | $256 \times 256$ | 12.0k | ImageNet (Deng et al., 2009) |
|  | Glide (Nichol et al., 2022) | $256 \times 256$ | 12.0k | ImageNet (Deng et al., 2009) |
|  | Midjourney (Midjourney, 2025) | $1024 \times 1024$ | 12.0k | ImageNet (Deng et al., 2009) |
|  | SD v1.4 (StableDiffusion, 2022) | $512 \times 512$ | 12.0k | ImageNet (Deng et al., 2009) |
|  | SD v1.5 (StableDiffusion, 2022) | $512 \times 512$ | 16.0k | ImageNet (Deng et al., 2009) |
|  | VQDM (Gu et al., 2022) | $256 \times 256$ | 12.0k | ImageNet (Deng et al., 2009) |
|  | Wukong (Wukong, 2023) | $512 \times 512$ | 12.0k | ImageNet (Deng et al., 2009) |

Table 12: Cross-model accuracy (Acc) performance on the Chameleon testset. For each training dataset, the first row indicates the Acc evaluated on the Chameleon testset, and the second row gives the Acc for "fake image/real image" for detailed analysis. All results of former methods can be sourced from the paper Yan et al. (2024a).

| Training Dataset | CNNSpot | FreDect | Fusing | UnivFD | DIRE | PatchCraft | NPR | AIDE | Ours |
|---|---|---|---|---|---|---|---|---|---|
| SD v1.4 | 60.11 | 56.86 | 57.07 | 55.62 | 59.71 | 56.32 | 58.13 | 62.60 | **73.44 (+10.84%)** |
|  | 8.86/98.63 | 1.37/98.57 | 0.00/99.96 | 17.65/93.50 | 11.86/95.67 | 3.07/96.35 | 2.43/100.00 | 20.33/94.38 | 65.47/79.44 |
| All GenImage | 60.89 | 57.22 | 57.09 | 60.42 | 57.83 | 55.70 | 57.81 | 65.77 | **79.10 (+13.33%)** |
|  | 9.86/99.25 | 0.89/99.55 | 0.02/99.98 | 85.52/41.56 | 2.09/99.73 | 1.39/96.52 | 1.68/100.00 | 26.80/95.06 | 76.77/82.20 |

results summarized in Table 12. When trained on the full GenImage dataset, HiDA-Net achieves an accuracy of **79.10%**, outperforming the previous best by a substantial margin of **13%**.

**Evaluation on HiRes-50K.** Next, we train all models on the full GenImage dataset and evaluate them on our proposed HiRes-50K benchmark. As shown in Table 13, our method outperforms all competing approaches. Notably, accuracy sometimes increases with resolution as richer cues expose synthetic artifacts, underscoring our method's scalability and robustness at extreme resolutions.

**Evaluation on GenImage.** For the public GenImage benchmark, models are trained on the SD v1.4 subset and tested across all official subsets. Detailed dataset statistics are provided in Table 11. The evaluation results are shown in Table. 14. This experiment verifies the model's consistency under varying generative styles and semantic contents.

**Evaluation on DRCT.** In addition, we evaluate performance on the DRCT dataset. All models are trained on the SD v1.4 subset of DRCT and tested across all available test sets. The results, presented in Table 15, demonstrate our model performs strongly, especially for high-resolution SDXL images at $1024 \times 1024$ and DR variants (partially inpainted images) showing the model's robustness under partially inpainted image conditions and domain-specific artifacts.

To summarize, our method consistently achieves strong performance across various benchmarks, particularly in high-resolution settings. Incorporating fake-real pairs via VAE reconstruction further boosts accuracy, confirming HiDA-Net's scalability and generalization.

## F    DETECTION OF LOCAL MANIPULATION ARTIFACTS

We introduce the *Token-wise Forgery Localization (TFL)* task to provide token-level supervision for localized manipulations and to enhance model interpretability.

As illustrated in Fig. 9, we adopt a **Random Patch Swap (RPS)** augmentation strategy. For a given pair of real and fake images, we randomly swap a portion of the corresponding regions to form a composite image that contains both authentic and manipulated content. When such paired samples are not available, we instead seclet a random real image and a random fake image for patch swapping. This process yields a

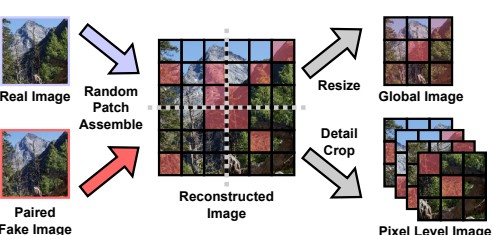

Figure 9: **Random Patch Swap (RPS) Augmentation**. Real and fake images (e.g., VAE-reconstructed pairs) are blended by swapping small local regions between them.

Table 13: Cross-model accuracy (Acc) performance on our HiRes-50K Dataset. All methods were trained on all available training subsets of GenImage and evaluated on HiRes-50K subsets with different resolutions.

| Resolution Range | [0, 900) | [900, 1200) | [1200, 1500) | [1500, 2000) | [2000, 2500) | [2500, 3000) | [3000, 5000) | [5000, ∞) | Avg |
|---|---|---|---|---|---|---|---|---|---|
| Image Numers | 1690 | 13330 | 12798 | 10524 | 7748 | 1142 | 2329 | 944 | 50568 |
| CNNSpot Wang et al. (2020) | 58.46 | 56.37 | 58.73 | 63.14 | 67.42 | 60.90 | 66.30 | 58.90 | 61.28 |
| FreDect Frank et al. (2020) | 63.67 | 55.63 | 57.31 | 58.90 | 58.16 | 67.78 | 57.23 | 51.06 | 59.72 |
| UnivFD Ojha et al. (2023) | 67.00 | 58.35 | 62.20 | 65.95 | 66.05 | 58.20 | 58.15 | 60.05 | 62.05 |
| DIRE Wang et al. (2023) | 59.11 | 64.25 | 62.10 | 66.66 | 75.84 | 63.40 | 69.31 | 62.18 | 65.36 |
| AIDE Yan et al. (2024a) | 65.87 | 57.29 | 49.90 | 58.23 | 51.88 | 65.31 | 61.04 | 42.16 | 56.46 |
| DRCT(UnivFD) Chen et al. (2024) | 57.81 | 61.67 | 62.26 | 62.26 | 62.31 | 59.72 | 58.74 | 46.40 | 58.90 |
| DRCT(ConvB) Chen et al. (2024) | 65.30 | 66.19 | 68.78 | 68.85 | 68.78 | 75.79 | 69.65 | 54.03 | 67.17 |
| TextureCrop(CNNDetect) | 57.45 | 52.81 | 59.83 | 63.17 | 65.31 | 55.56 | 70.65 | 69.49 | 60.67 |
| TextureCrop(UnivFD) | 64.14 | 58.90 | 61.16 | 62.82 | 66.48 | 59.32 | 59.11 | 66.30 | 62.28 |
| Ours | **82.98** | **81.39** | **78.16** | **78.99** | **88.26** | **80.18** | **82.88** | **69.84** | **80.33 (+13.16%)** |

Table 14: Cross-dataset accuracy on the GenImage Dataset. All methods are trained on SDv1.4 and evaluated on all GenImage.

| Method | Time | Midjourney | SDv1.4 | SDv1.5 | ADM | GLIDE | Wukong | VQDM | BigGAN | Avg |
|---|---|---|---|---|---|---|---|---|---|---|
| ResNet-50 He et al. (2016) | CVPR2016 | 54.9 | 99.9 | 99.7 | 53.5 | 61.9 | 98.2 | 56.6 | 52.0 | 72.1 |
| DeiT-S Touvron et al. (2021) | ICML2021 | 55.6 | 99.9 | 99.8 | 49.8 | 58.1 | 98.9 | 56.9 | 53.5 | 71.6 |
| Swin-T Liu et al. (2021) | ICCV2021 | 62.1 | 99.9 | 99.8 | 49.8 | 67.6 | 99.1 | 62.3 | 57.6 | 74.8 |
| CNNSpot Wang et al. (2020) | CVPR2020 | 52.8 | 96.3 | 95.9 | 50.1 | 39.8 | 78.6 | 53.4 | 46.8 | 64.2 |
| Spec Zhang et al. (2019) | WIFS2019 | 52.0 | 99.4 | 99.2 | 49.7 | 49.8 | 94.8 | 55.6 | 49.8 | 68.8 |
| F3Net Qian et al. (2020) | ECCV2020 | 50.1 | 99.9 | 99.9 | 49.9 | 50.0 | 99.9 | 49.9 | 49.9 | 68.7 |
| GramNet Liu et al. (2020) | CVPR2020 | 54.2 | 99.2 | 99.1 | 50.3 | 54.6 | 98.9 | 50.8 | 51.7 | 69.9 |
| UnivFD Ojha et al. (2023) | CVPR2023 | 93.9 | 96.4 | 96.2 | 71.9 | 85.4 | 94.3 | 81.6 | 90.5 | 88.8 |
| PatchCraft Zhong et al. (2023a) | Arxiv | 79.0 | 89.5 | 89.3 | 77.3 | 78.4 | 89.3 | 83.7 | 72.4 | 82.3 |
| NPR Tan et al. (2024b) | CVPR2024 | 81.0 | 98.2 | 97.9 | 76.9 | 89.8 | 96.9 | 84.1 | 84.2 | 88.6 |
| FreqNet Tan et al. (2024a) | AAAI2024 | 89.6 | 98.8 | 98.6 | 66.8 | 86.5 | 97.3 | 75.8 | 81.4 | 86.8 |
| FatFormer Liu et al. (2024) | CVPR2024 | 92.7 | **100.0** | 99.9 | 75.9 | 88.0 | 99.9 | 98.8 | 55.8 | 88.9 |
| DRCT Chen et al. (2024) | ICML2024 | 91.5 | 95.0 | 94.4 | 79.4 | 89.1 | 94.6 | 90.0 | 81.6 | 89.4 |
| AIDE Yan et al. (2024a) | ICLR2025 | 79.4 | 99.7 | 99.8 | 78.5 | 91.8 | 98.7 | 80.3 | 66.9 | 86.8 |
| Effort Yan et al. (2024b) | ICML2025 | 82.4 | 99.8 | 99.8 | 78.7 | 93.3 | 97.4 | 91.7 | 77.6 | 91.1 |
| SAFE Li et al. (2024) | KDD2025 | 95.3 | 99.4 | 99.3 | 82.1 | 96.3 | 98.2 | 96.3 | 97.8 | 95.6 |
| C2P-CLIP Tan et al. (2025) | AAAI2025 | 88.2 | 90.9 | 97.9 | 96.4 | 99.0 | 98.8 | 96.5 | 98.7 | 95.8 |
| Ours/No VAE Augmentation | - | **97.8** | 98.4 | 98.3 | 86.2 | 98.0 | 98.4 | 95.6 | 96.2 | **96.1 (+0.3%)** |
| Ours/SDv1.4 VAE | - | 94.2 | 99.1 | 99.2 | 92.7 | **99.1** | 98.0 | 96.9 | 97.8 | **97.1 (+1.3%)** |

continuous-valued "soft" supervision label $y_{\text{token}} \in [0, 1]$ for each ViT patch token, calculated as the average of binary pixel-level labels within the corresponding patch area.

We present prediction results from our HiDA-Net model on partially manipulated images in Fig. 10. This image are generated via AI-based inpainting using Stable Diffusion v1.4. Specifically, we partially overwrite regions in the real image by adding noise to its latent representations, followed by reconstruction through the diffusion process. The resulting synthetic regions are then blended with the original image using a predefined binary mask to obtain partially forged samples.

Our model accurately localizes the manipulated regions at the tile level. As shown in the prediction maps, regions with higher mask intensity (brighter areas) correspond to higher predicted likelihoods of being fake. This demonstrates the model's fine-grained localization capability and its sensitivity to subtle, localized modifications.

## G  MORE ABLATION STUDY

We conduct a comprehensive ablation study to systematically evaluate the impact of varying the number of cropped image tiles that are fed into the **FAM** module during both the training and inference stages. The evaluation protocol follows the same experimental settings as those used in the comparisons on the *GenImage* benchmark, ensuring the results are directly comparable. As illustrated in Table 16, the findings reveal a clear and consistent trend: increasing the number of local tiles leads to noticeable improvements in detection performance. This improvement can be attributed to the model's enhanced ability to capture more detailed and spatially diverse cues as additional tiles are introduced.

Notably, the *full-coverage configuration*, in which the **FAM** module receives the maximum number of non-overlapping high-frequency tiles extracted from the image, achieves the highest accuracy among all tested settings. This outcome provides strong evidence that incorporating fine-grained local information is highly beneficial for manipulation detection. In particular, by aggregating and integrating localized features across multiple regions, the model is able to construct a richer and more

| Method | SD Variants | | | | | | Turbo Variants | | LCM Variants | | ControlNet Variants | | | DR Variants | | | Avg |
|---|---|---|---|---|---|---|---|---|---|---|---|---|---|---|---|---|---|
| | LDM | SDv1.4 | SDv1.5 | SDv2 | SDXL | SDXL-Refiner | SD-Turbo | SDXL-Turbo | LCM-SDv1.5 | LCM-SDXL | SDv1-Ctrl | SDv2-Ctrl | SDXL-Ctrl | SDv1-DR | SDv2-DR | SDXL-DR | |
| CNNSpot Wang et al. (2020) | 99.87 | 99.91 | 99.90 | 97.55 | 66.25 | 86.55 | 86.15 | 72.42 | 98.26 | 61.72 | 97.96 | 85.89 | 82.84 | 60.93 | 51.41 | 50.28 | 81.12 |
| F3Net Qian et al. (2020) | 99.85 | 99.78 | 99.79 | 88.66 | 55.85 | 87.37 | 68.29 | 63.66 | 97.39 | 54.98 | 97.98 | 72.39 | 81.99 | 65.42 | 50.39 | 50.27 | 77.13 |
| CLIP/RN50 Radford et al. (2021) | 99.00 | 99.99 | 99.96 | 94.61 | 62.08 | 91.43 | 83.57 | 64.40 | 98.97 | 57.43 | 99.74 | 80.69 | 82.03 | 65.83 | 50.67 | 50.47 | 80.05 |
| GramNet Liu et al. (2020) | 99.40 | 99.01 | 98.84 | 95.30 | 62.63 | 80.68 | 71.19 | 69.32 | 93.05 | 57.02 | 89.97 | 75.55 | 82.68 | 51.23 | 50.01 | 50.08 | 76.62 |
| De-fake Sha et al. (2023) | 92.10 | 99.53 | 99.51 | 89.65 | 64.02 | 69.24 | 92.00 | 93.93 | 99.13 | 70.89 | 58.98 | 62.34 | 66.66 | 50.12 | 50.16 | 50.00 | 75.52 |
| Conv-B Liu et al. (2022) | 99.97 | 100.0 | 99.97 | 95.84 | 64.44 | 82.00 | 80.82 | 60.75 | 99.27 | 62.33 | 99.80 | 83.40 | 73.28 | 61.65 | 51.79 | 50.41 | 79.11 |
| UnivFD Ojha et al. (2023) | 98.30 | 96.22 | 96.33 | 93.83 | 91.01 | 93.91 | 86.38 | 85.92 | 90.44 | 88.99 | 90.41 | 81.06 | 89.06 | 51.96 | 51.03 | 50.46 | 83.46 |
| DIRE/SDv1 Wang et al. (2023) | 98.19 | 99.94 | 99.96 | 68.16 | 53.84 | 71.93 | 58.87 | 54.35 | 99.78 | 59.73 | 99.65 | 64.20 | 59.13 | 51.99 | 50.04 | 49.97 | 71.23 |
| DIRE/SDv2 Wang et al. (2023) | 54.62 | 75.89 | 76.04 | 99.87 | 59.90 | 93.08 | 99.77 | 57.55 | 87.29 | 72.53 | 67.85 | 99.69 | 64.40 | 49.96 | 52.48 | 49.92 | 72.55 |
| DRCT/SDv1.4 Chen et al. (2024) | 99.91 | 99.90 | 99.90 | 96.32 | 83.87 | 85.63 | 91.88 | 70.04 | 99.66 | 78.76 | 99.90 | 95.01 | 81.21 | 99.90 | 95.40 | 75.39 | 90.79 |
| DRCT/SDv2 Chen et al. (2024) | 99.66 | 98.56 | 98.48 | 99.85 | 96.10 | 98.68 | 99.59 | 83.30 | 98.45 | 93.78 | 96.68 | 99.85 | 97.66 | 93.91 | 99.87 | 90.39 | 96.55 |
| Ours/No VAE Augmentation | 98.66 | 98.75 | 98.75 | 98.73 | 98.76 | 98.75 | 97.74 | 97.56 | 98.55 | 98.75 | 98.75 | 98.37 | 98.15 | 90.36 | 74.78 | 71.17 | 94.79 (-1.76%) |
| Ours/SDv1.4 VAE | 98.82 | 98.91 | 98.90 | 97.95 | 98.99 | 98.84 | 98.45 | 98.82 | 98.40 | 98.89 | 98.52 | 98.65 | 98.44 | 98.99 | 97.38 | 94.91 | 98.37 (+1.82%) |

Table 15: Cross-model accuracy (Acc) performance on the different testing subsets of DRCT. Methods are trained on the SDv1.4 subset of DRCT. Results of former methods can be sourced from the paper DRCT Chen et al. (2024).

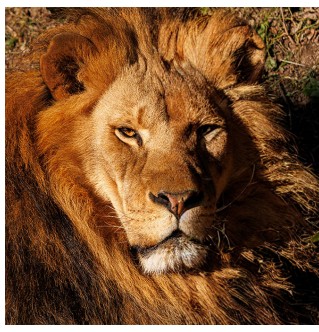

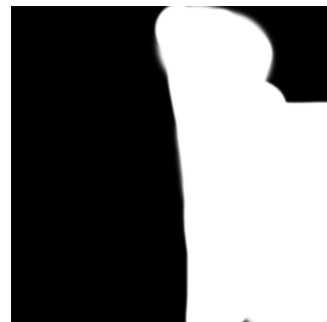

(a) AI-Inpainted Image  (b) Predicted Inpainting Mask  (c) GT Inpainting Mask

Figure 10: Visual comparison between the AI-inpainted image, the predicted inpainting mask, and the ground-truth mask.

discriminative representation, thereby significantly enhancing its robustness and overall detection effectiveness.

Furthermore, when the FAM module is executed without incorporating any local tile inputs and instead depends solely on the information provided by the global branch, the overall detection accuracy on the *GenImage* benchmark experiences a notable decline, dropping to **84.24%**. This considerable reduction clearly demonstrates that global features alone are insufficient for achieving optimal performance, and it further emphasizes the crucial role that localized visual cues play in strengthening the model's detection capability. In other words, fine-grained local details extracted from image tiles are indispensable for complementing global representations and ensuring robust performance.

To gain a deeper understanding of how the number of cropped tiles influences model effectiveness, we conduct a systematic investigation into the relationship between tile quantity and detection performance. Specifically, we evaluate the impact of different tile sampling strategies on both our *HiRes-50K* dataset and the *GenImage* benchmark. In this study, we fix the number of tiles $n$ to one of the following values: $\{1, 2, 4, 8, 16\}$. When $n = 1$, the model is restricted to using only a single center crop during inference, which provides a limited view of the image. For larger values of $n$, i.e., when $n > 1$, multiple tiles are randomly sampled, and this sampling process is consistently applied during both the training and inference stages.

In contrast to these fixed strategies, our proposed **FAM module** employs a more flexible and adaptive approach. During training, instead of adhering to a predetermined number of tiles, it dynamically samples a variable number ranging from 1 to 16. This stochastic sampling not only introduces greater data diversity but also improves the model's ability to generalize. At inference time, rather than relying on random selection, the module adopts a *full-coverage tile cropping strategy*, ensuring that all relevant local regions are exhaustively considered. This combination of dynamic sampling in training and comprehensive coverage during inference represents a key factor in the effectiveness of the FAM module.

Table 17 presents the experimental results obtained on the *GenImage* benchmark, where all models are trained using the complete training set and subsequently evaluated under different tile configurations during inference. From the reported results, we observe a clear and consistent trend: as the number of cropped tiles increases, the detection performance steadily improves. Specifically, when

Table 16: Ablation study on the number of crop tiles in FAM.

| Tile Nums | 1 | 2 | 4 | 8 | 16 | FAM |
|---|---|---|---|---|---|---|
| ACC (%) | 92.14 | 93.34 | 95.63 | 95.69 | 95.89 | **96.10** |

Table 17: Cross-model accuracy (Acc) performance on our HiRes-50K Dataset with different tile crop strategies.

| Resolution Range | $[0, 900)$ | $[900, 1200)$ | $[1200, 1500)$ | $[1500, 2000)$ | $[2000, 2500)$ | $[2500, 3000)$ | $[3000, 5000)$ | $[5000, \infty)$ | Avg |
|---|---|---|---|---|---|---|---|---|---|
| Image Numbers | 1690 | 13330 | 12798 | 10524 | 7748 | 1142 | 2329 | 944 | 50568 |
| 1 Crop Tile | 75.3 | 74.9 | 72.8 | 75.1 | 81.6 | 73.7 | 77.2 | 62.1 | 74.09 |
| 2 Crop Tiles | 79.9 | 81.4 | 76.7 | 76.0 | 88.4 | 74.7 | 79.4 | 66.8 | 77.91 |
| 4 Crop Tiles | 83.4 | 82.1 | 78.8 | 78.9 | 88.6 | 78.1 | 80.2 | 68.1 | 79.78 |
| 8 Crop Tiles | 83.4 | 83.4 | 79.5 | 78.4 | 89.1 | 78.3 | 81.2 | 68.7 | 80.25 |
| 16 Crop Tiles | 83.6 | 84.3 | 79.5 | 78.0 | 89.5 | 77.2 | 82.1 | 67.7 | 80.24 |
| Ours FAM | 82.98 | 81.39 | 78.16 | 78.99 | 88.26 | 80.18 | 82.88 | 69.8 | 80.33 |

the model is restricted to a single tile, the accuracy reaches only **74.09%**. However, as the number of tiles is gradually increased, the model is able to capture a richer set of localized features, leading to progressively better performance. At the maximum setting of $n = 16$ tiles, the accuracy improves to **80.33%**, corresponding to a relative gain of more than six percentage points compared to the single-tile baseline. These findings highlight the importance of incorporating multiple localized views of an image, which provide complementary information to global representations and thereby enhance the robustness and effectiveness of the detection model.

Table 18: Intra-Dataset accuracy (Acc) performance on our HiRes-50K Dataset with different test-time tile crop strategies.

| Resolution Range | $[0, 900)$ | $[900, 1200)$ | $[1200, 1500)$ | $[1500, 2000)$ | $[2000, 2500)$ | $[2500, 3000)$ | $[3000, 5000)$ | $[5000, \infty)$ | Avg |
|---|---|---|---|---|---|---|---|---|---|
| Image Numbers | 1690 | 13330 | 12798 | 10524 | 7748 | 1142 | 2329 | 944 | 50568 |
| 1 Crop Tile | 97.8 | 97.6 | 98.2 | 98.1 | 97.7 | 97.8 | 96.4 | 94.3 | 97.24 |
| 2 Crop Tiles | 98.3 | 97.2 | 98.9 | 98.2 | 98.5 | 97.7 | 97.7 | 95.2 | 97.71 |
| 4 Crop Tiles | 98.2 | 98.1 | 98.7 | 99.0 | 98.7 | 97.9 | 97.6 | 95.7 | 97.99 |
| 8 Crop Tiles | 98.4 | 98.3 | 99.0 | 98.6 | 98.7 | 98.3 | 98.2 | 96.0 | 98.19 |
| 16 Crop Tiles | 98.4 | 98.4 | 99.1 | 98.7 | 98.7 | 98.6 | 98.1 | 96.3 | 98.29 |
| Ours FAM | 98.5 | 99.2 | 97.8 | 99.0 | 99.1 | 99.5 | 97.3 | 96.0 | 98.30 |

Table 18 shows results on the HiRes-50K dataset, where we split the data into 50% for training, 25% for validation, and 25% for testing. A similar trend is observed: increasing the number of tiles used during inference leads to clear performance improvements.

Overall, performance improves as more random crops are utilized, demonstrating the importance of diverse spatial coverage. Our FAM-based strategy consistently achieves the highest accuracy, highlighting the effectiveness of its adaptive sampling mechanism.

## H  POTENTIAL SOCIETAL IMPACTS

The release of the HiRes-50K dataset introduces both valuable opportunities and important responsibilities. As one large-scale, high-resolution benchmark for AI-generated image detection, it enables rigorous evaluation under realistic visual conditions and fosters the development of more robust forensic tools. This can greatly assist in identifying synthetic media in journalism, legal investigations, and content moderation. However, by making such data widely available, there is also the risk that generative model developers may exploit it to enhance the realism of forgeries and evade detection. To mitigate misuse, we encourage responsible dataset use aligned with research ethics and advocate for transparent licensing, access control, and ongoing dialogue with policymakers and interdisciplinary stakeholders.

## I  DETAILS ON LARGE LANGUAGE MODEL USAGE

In this work, a Large Language Model (LLM) was used only as an auxiliary tool to polish writing, improve grammar, and refine expressions for clarity. It was not used to generate scientific content, design experiments, analyze results, or write substantive parts of the paper. All conceptual ideas, technical contributions, and experiment analyses were conducted entirely by the authors, with the LLM serving solely as a language aid rather than a content creator.

