# OpenReview forum: "No Pixel Left Behind: A Detail-Preserving Architecture for Robust High-Resolution AI-Generated Image Detection"
_ICLR.cc/2026/Conference — ICLR 2026 Poster_

### Official Review · Reviewer_MJAJ · 2025-10-30

**Soundness:** 3
**Presentation:** 3
**Contribution:** 3
**Rating:** 4
**Confidence:** 3

**Summary:**

This paper presents HiDA-Net, a high-resolution AI-generated image detection framework designed to retain fine-grained details by aggregating information from full-resolution local tiles and a global downsampled view. The model integrates Token-wise Forgery Localization (TFL) and JPEG Quality Factor Estimation (QFE). The authors further introduce HiRes-50K dataset. Experimental results demonstrate that HiDA-Net achieves state-of-the-art performance.

**Strengths:**

1. Structure of Paper; The paper is technically sound and well-executed, with comprehensive ablation studies that explore the effects of different loss functions, numbers of tiles, and other architectural choices. These analyses effectively demonstrate the contribution of each component to the overall performance. The proposed framework is both conceptually clear and empirically validated.

**Weaknesses:**

1. Visual Analysis on process: The paper could provide a deeper analysis of the interaction between local detail tokens and global tokens. For example, it would be valuable to examine HiDA-Net’s behavior on AI-inpainted images, where some tiles contain real content while others contain generated regions. Understanding how local detail tokens respond in such mixed cases and how the local aggregator fuses these heterogeneous signals would offer stronger insight into the model’s robustness and interpretability.

**Questions:**

1. Since tiling is a core aspect of the method, a more comprehensive ablation would strengthen the contribution. Could the authors explore overlapping tile strategies or compare padding-based approaches instead of simply resizing patches smaller than 224 pixels?

2. Is the CLS token for local detail tokens rotation-invariant? For instance, would identical patches with a 90° rotation produce the same output representation ($t_{out}$)?

3. How are the ratios (weights) among different loss functions determined?

---

> ### Author Response · Authors · 2025-11-23
>
> We thank Reviewer MJAJ for the thoughtful comments and suggestions. We have revised the paper accordingly and respond point-by-point below.
>
> 1. **Visual Analysis of Local Tokens.** We have a qualitative study (see **Figure 10** in Appendix F) showing visual comparisons between AI-inpainted images, predicted masks, and ground-truth regions. In partially edited cases, local tiles within the manipulated region produce clearly higher forgery scores, and the Local Aggregator fuses these heterogeneous signals into a global decision that highlights the forged area. The visualization illustrates that HiDA-Net effectively exploits local detail tokens by showing their distinct response in such mixed real/fake cases.
>
> 2. **Tiling Strategy Ablation.** We experimented with an overlapping tiling strategy where a second grid of tiles is introduced, centered at the intersections of the original, resulting in a half-tile overlap in both dimensions. Under this setting, the accuracy changes only marginally (e.g., 96.11% original vs. 96.08% overlapped on Genimage), indicating that adding overlapping tiles bring little benefit relative to their added complexity. For patches smaller than $224\times224$, our focus is on genuinely high-resolution inputs, and available test data for smaller than $224\times224$ is limited; therefore, we opt to avoid additional padding-based variants and retain the original tiling strategy as a practical and effective choice.
>
> 3. **Rotation Invariance.** Because our backbone is a ViT, its CLS token is not inherently rotation-invariant by design. To assess the impact of this property, we conduct an additional experiment on the GenImage dataset, rotating each image by $0^\circ, 90^\circ, 180^\circ$ and $270^\circ$. The resulting accuracies are reported in the table below. Although the learned representations are not mathematically rotation-invariant, HiDA-Net remains highly robust under these rotations, exhibiting only minimal performance degradation.
>
>    | Rotation     | 0      | 90     | 180    | 270    |
>    | ------------ | ------ | ------ | ------ | ------ |
>    | Accuracy (%) | 96.11% | 95.74% | 95.85% | 95.60% |
>
> 4. **Loss Weights.** In practice, the three loss is on the same numerical scale. We clarified in Section 4.4 that we utilize equal weighting ($\alpha=\beta=1$) for all loss components, as this simple configuration yields stable convergence and strong performance without complex tuning.

---

### Official Review · Reviewer_LuBd · 2025-10-30

**Soundness:** 3
**Presentation:** 3
**Contribution:** 2
**Rating:** 6
**Confidence:** 3

**Summary:**

The paper proposes HiDA-Net and HiRes-50K benchmark to prevent performance drops of AIDI on high-resolution images brought by detail loss and limited generalization. Specifically, HiDA-Net aggregates multiple information (local details, global structure, JPEG) with FAM to provide a better understanding of the AIGC image. Also, the introduces HiRes-50K to enable realistic high-resolution evaluation under varied resolution and image quality. Overall, the HiDA-Net achieves SoTA performance in the experimental comparison.

**Strengths:**

- The motivation is clear and intuitive. The detailed information loss will introduce misunderstanding for the detection model, especially for the AIGC scene, where most generated models produce well-structured but detail-failed images.
- The model design is reasonable and theoretically proven.
- The experiment is comprehensive to demonstrate the effectiveness of the proposed network and benchmarks.

**Weaknesses:**

- The open-source AIGC image dataset and real-image dataset are innumerable. The paper does not sufficiently explain the core basis for HiRes-50K to surpass existing data resources in terms of irreplaceability or value increment.
- The experiments primarily rely on outdated models (e.g., SD v1.4, SDXL) for generating AI-synthesized images with limited coverage of other mainstream high-resolution generative models—especially advanced models updated after 2024 (e.g., SD3.5, FLUX, Qwen-Image).
- The analysis on input degradation focuses on JPEG artifacts; however, the real-world imaging encounters noise, blur, defocus, and moire patterns. Extensive studies on more complicated degradation would improve the integrity of the discussion.
-  More recent model have employed multiple strategies, e.g., RL and adversarial learning to improve their capability of "actively evading detection". The experimental design of the paper does not cover such scenarios.

**Questions:**

See weakness.

---

> ### Author Response · Authors · 2025-11-23
>
> We thank Reviewer LuBd for the constructive and insightful comments. We have revised the paper accordingly and respond to each concern below.
>
> 1. **Value of HiRes-50K.** HiRes-50K offers unique value by bridging the gap between academic benchmarks and real-world deployment. Built from "in-the-wild" platforms and manually curated for "human-hard" quality, it represents a distribution far closer to reality than synthetic datasets derived from fixed prompts. Because the images are user-generated, they encompass diverse pipelines, which includes local editing, inpainting, and super-resolution and span a significantly broader resolution range. Unlike existing datasets clustered around ~1MP, a substantial portion of HiRes-50K ranges from 2 megapixels up to 64 megapixels, significantly extending the evaluation boundary for high-resolution detection. We have updated Section 5 to highlight these distinctive contributions, and Appendix C provides a detailed comparison further demonstrating our dataset's superior resolution coverage relative to existing benchmarks.
> 2. **Coverage of Newer Models.** To ensure our evaluation remains current, we have expanded our experiments to include images generated by recent state-of-the-art models, including Midjourney v6.1, SD3, and FLUX (**Table 5**), comparing HiDA-Net against newly proposed detectors (**Table 2 & 5**). Furthermore, since HiRes-50K is sourced from active online communities rather than a static offline engine, it inherently covers an evolving landscape of generators, including close source commercial systems released after 2024. This design ensures the benchmark remains relevant as technology advances, and our experiments confirm HiDA-Net’s robust generalization to these emerging sources.
> 3. **Degradation Analysis Beyond JPEG.** While the QFE module specifically targets compression, our robustness analysis includes resizing and blur, and extends to Gaussian noise (as shown in Table 8 of the revised manuscript). These experiments confirm that HiDA-Net maintains strong performance across the most frequent distortions found in online media pipelines, ensuring reliability beyond just compression artifacts.
> 4. **Adversarial Evasion.** We clarify that this work focuses on the robust detection of non-adaptive high-resolution forgeries. While detection-evading generators (e.g., those trained via RL or adversarial attacks) represent a significant threat, they constitute a distinct adversarial setting orthogonal to the scope of this paper. We have acknowledged this as a critical direction for future research.

---

### Official Review · Reviewer_L8Yu · 2025-10-31

**Soundness:** 2
**Presentation:** 2
**Contribution:** 2
**Rating:** 4
**Confidence:** 3

**Summary:**

This paper proposes a High-Resolution Detail-Aggregation Network (HiDA-Net) for detecting AI-generated images in high-resolution inputs. To effectively capture both global context and local details, HiDA-Net processes the input image through two complementary paths — a global path and a local path — where global and local features are extracted for classification.

To aggregate these features effectively, the authors introduce three key modules: the Feature Aggregation Module (FAM), Token-wise Forgery Localization (TFL), and JPEG Quality Factor Estimation (QFE). Specifically, FAM aggregates the [CLS] tokens from local detail tiles and the global token to form a discriminative feature representation for classification. TFL introduces a Random Patch Swap (RPS) augmentation to create partially manipulated images, providing token-level supervision that enhances spatial sensitivity to localized forgeries. QFE, implemented as a lightweight MLP, predicts the JPEG quality factor to help disentangle compression artifacts from generative ones.

In addition, the paper presents a new dataset, HiRes-50K, which provides a large number of high-quality images for evaluating detection performance. Experimental results show that the proposed method achieves superior performance on both the Chameleon and HiRes-50K datasets. However, several questions and concerns remain, as listed below:

**Strengths:**

1. The paper convincingly identifies and quantifies how resizing harms detection by losing high-frequency details.
2. It introduces a new and high-quality dataset, named HiRes-50K, for evaluation.

**Weaknesses:**

1. The proposed method includes both a global and a local path, but uses the same transformer blocks to process global and local images. How do the proposed FAM, TFL, and QFE modules adaptively extract and distinguish local and global features for classification? It is unclear why these modules can effectively capture both types of features simultaneously.

2. The computational complexity of transformer blocks is quadratic. When more local patches are used, this inevitably increases the processing time and computational cost. How is this issue addressed or mitigated?

3. The paper claims that local details are crucial for detection performance, implying that the size of the local image patches plays an important role. How does the patch size affect overall performance?

4. Important ablation studies are missing. Since the proposed method includes both a global and a local path, an ablation study should be provided to evaluate the effectiveness of this dual-path design. How do the global and local paths individually contribute to the final performance?

5. For a fair comparison, the paper should clearly provide implementation details, including the training dataset and the model complexity (e.g., number of parameters, FLOPs).

**Questions:**

The questions I concern have been posted in the Weakness section. The authors can prepare their rebuttal with reference to the review comments in Weakness section.

---

> ### Author Response · Authors · 2025-11-23
>
> We thank Reviewer L8Yu for the careful reading of our manuscript and the constructive comments. We have revised the paper accordingly and address each point in detail below. Line and figure numbers refer to the revised version.
>
> 1. **Dual-Path Design and Feature Adaptation.** Although the global and local images are processed by the same ViT backbone, their representations are handled differently afterward: in FAM, we first refine global and local [CLS] tokens with a shared lightweight 2-layer transformer, but only the local [CLS] tokens are further processed by a dedicated Local Aggregator transformer to obtain feature representations distinct from the global one, while the global [CLS] token is kept as a coarse, scene-level summary. TFL is shared by the global and local branches and is designed to enhance the feature extraction capacity of this shared transformer via token-level supervision. QFE is applied exclusively on local features, since JPEG artifacts are mostly manifested in high-frequency regions, which further encourages the local branch to disentangle compression artifacts from generative ones. Together, these design choices make the global branch provide robust global semantics, while the local branch is explicitly driven to capture fine-grained, high-resolution forensic cues. For the ablation study on global and local paths, please refer to Point 4.
>
> 2. **Computational Complexity.** We emphasize that our tiling strategy ensures computational cost scales linearly, rather than quadratically, with input resolution. By processing fixed-size tiles independently, the quadratic complexity of self-attention is confined within each tile. The subsequent cross-tile attention operates solely on [CLS] tokens, introducing negligible overhead. As demonstrated in **Figure 6** of the revised manuscript, the FLOPs increase linearly with the number of tiles, making HiDA-Net significantly more efficient for high-resolution inputs than standard full-image attention mechanisms.
>
> 3. **Tile Size Selection.** We selected a $224\times224$ tile size to match the native resolution of standard pre-trained backbones, ensuring optimal feature extraction without information loss. Experiments with a $336\times336$ variant yielded lower performance (95.00% vs. 96.11%), likely due to suboptimal pre-training at that resolution.
>
>    | Backbone Model  | CLIP ViT-L/14 | CLIP ViT-L/14@336px |
>    | --------------- | ------------- | ------------------- |
>    | Acc on Genimage | 96.11         | 95.00               |
>
> 4. **Ablation of Global vs. Local Paths.** Ablation of Global vs. Local Paths. We have added comprehensive ablation studies to validate the necessity of our dual-path design. **Table 9** (in the revision) confirms this on the GenImage dataset, while the table below details the performance breakdown on HiRes-50K. As shown, the local path is the primary performance driver in high-resolution scenarios (77.91% vs. 66.92% for global only), confirming that local details are critical. However, completely removing the global path degrades performance on general benchmarks (e.g., GenImage), indicating that global semantics provide essential context. The combined model achieves the highest accuracy (80.33%), proving that fusing multi-scale information is optimal. Furthermore, Table 17 in Appendix G systematically demonstrates that increasing the number of tiles leads to consistent performance gains, validating our full-coverage strategy.
>
>    | Resolution Range | 0-900 | 900-1200 | 1200-1500 | 1500-2000 | 2000-2500 | 2500-3000 | 3000-5000 | >5000 | Avg   |
>    | ---------------- | ----- | -------- | --------- | --------- | --------- | --------- | --------- | ----- | ----- |
>    | Only Local       | 73.89 | 79.78    | 81.05     | 82.80     | 86.94     | 82.80     | 77.71     | 58.28 | 77.91 |
>    | Only Global      | 68.31 | 65.92    | 67.36     | 71.34     | 79.46     | 65.29     | 62.10     | 55.57 | 66.92 |
>    | Both             | 82.98 | 81.39    | 78.16     | 78.99     | 88.26     | 80.18     | 82.88     | 69.84 | 80.33 |
>
> 5. **Model Information.**  We have listed the training datasets used for each experiment. We now report the model parameters size and compute complexity in section 6.4.

---

### Official Review · Reviewer_Uopj · 2025-10-31

**Soundness:** 3
**Presentation:** 2
**Contribution:** 3
**Rating:** 6
**Confidence:** 4

**Summary:**

The paper addresses the challenge of detecting high-resolution AI-generated images. The proposed framework HiDA-Net processes images in original resolution enabling increased quality of predictions and achieves SOTA performance. The paper also proposes a new dataset HiRes-50K that consists of >50k images of high resolution.

**Strengths:**

1. The motivation for the method is absolutely clear and supplemented with math and illustrations.
2. The proposed method achieves SOTA performance on several datasets.
3. The paper proposed a novel high-resolution HiRes-50K dataset that may be valuable for the community.
4. The paper includes extensive ablation on the proposed method.

**Weaknesses:**

Major weaknesses:
1. The proposed method is not compared with recent AI-generated image detection methods, like [1 - 3].
2. The proposed method has an increased inference time for high resolution images compared to the other approaches. But what is the difference in speed between the HiRes-50K and the other methods on standard resolutions, like 224 $\times$ 224?
3. I have not found the explicit list of models that are used in creating the HiRes-50K dataset. It is important to include the relevant models in the dataset.
4. Lines 333-334 describe the resizing applied to the real part of the proposed dataset. What was the distribution of the original sizes of it? The resizing operations on the real part of the dataset may have led to models learning resizing artifacts instead of the synthetic images signs.

[1] Karageorgiou, Dimitrios, et al. "Any-resolution ai-generated image detection by spectral learning," Proceedings of the Computer Vision and Pattern Recognition Conference. 2025.
[2] Koutlis, Christos, and Symeon Papadopoulos. "Leveraging representations from intermediate encoder-blocks for synthetic image detection," European Conference on Computer Vision, 2024.
[3] Guillaro, Fabrizio, et al. "A bias-free training paradigm for more general ai-generated image detection." Proceedings of the Computer Vision and Pattern Recognition Conference. 2025.

Minor weaknesses:
1. The quotes could be isolated from the text to improve readability (e.g. “generative models (Karras et al. (2017))”)
2. The text mentions that the dataset contains only images with resolutions below 1K to over 10K pixels, however Figure 4 shows that the biggest resolution is below 6K. Does the dataset contain higher resolutions and what is the percentage of it compared to the other dataset?

**Questions:**

The authors should clarify issues described in the Weaknesses section.

---

> ### Author Response · Authors · 2025-11-23
>
> We thank Reviewer Uopj for the careful reading of our manuscript and the constructive comments. We have revised the paper accordingly and address each point in detail below. Line and figure numbers refer to the revised version.
>
> 1. **Comparison with Recent Methods.** We have added direct comparisons with SPAI [1] and RINE [2] in **Table 5** and **Table 2** of the revised manuscript. In **Table 5** which adopt the mixed test set from SPAI, our HiDA-Net achieves performance comparable to SPAI and significantly outperforms RINE.  In **Table 2**, we evaluate these methods in high-resolution scenarios of our HiRes-50K dataset, HiDA-Net outperforms SPAI by approximately **8%** and and RINE by **18%**, highlighting our advantage in processing high-resolution inputs. Due to B-Free [3] releasing only inference code, we are unable to conduct a fair re-training and evaluation, but we have discussed its paradigm and relation to our work in more detail in the revised paper.
> 2. **Inference Computational Cost.** We explicitly quantified computational costs in **Section 6.4** and **Figure 6**.  For Standard Resolution ($224\times224$), HiDA-Net requires roughly $2\times$ the FLOPs of the ViT backbone (processing 1 global + 1 local view). For High Resolution, FLOPs increase linearly with the number of tiles (pixels). This linear scaling is far more efficient than quadratic attention mechanisms over full images.
> 3. **HiRes-50K Data Sources.** HiRes-50K aggregates "in-the-wild" images to evaluate realistic deployment scenarios. As detailed in **Section 5**, images are sourced from platforms like Civitai and LiblibAI. Many synthetic images come from user-uploaded content whose metadata and exact generation pipelines are missing or incomplete. But what is certain is that images in our dataset cover a diverse mixture of models (Midjourney, SD variants, proprietary APIs), and some images include post-processing pipelines (inpainting, upscaling), rather than relying on a fixed list of single-source generators. We believe this diversity makes HiRes-50K not only substantially more challenging but also far more representative of the real distribution of AIGI images found on the Internet, thereby providing a more faithful reflection of a model’s practical performance.
> 4. **Real Image Resizing.** To prevent the model from learning upsampling artifacts, we strictly downsample high-quality real images (>16MP) using Lanczos interpolation to match the synthetic distribution. Which reflects how real high-resolution photos are typically resized when shared online and avoids introducing upsampling artifacts.
> 5. **Minor Corrections.**
>    - **Formatting:** Citations and quotes have been improved for readability.
>    - **Figure 4:** We corrected the plotting issue; the figure now correctly reflects the resolution distribution up to 10K pixels.

---

### Author Response · Authors · 2025-12-01
**Summary of Reviews, Rebuttals, and Revisions**

We thank all the reviewers for their constructive feedback. We have actively engaged with all reviewers, resulting in a consensus on the paper's strengths and a significantly improved manuscript. Below, we summarize the key aspects.

# **1. Consensus on Strengths**

The reviewers unanimously recognized the value of our work, highlighting four core aspects:

- **Theoretical Foundation and Motivation:** **L8Yu** commended the convincing quantification of the core resizing problem. **Uopj** noted the motivation was supplemented by mathematical proofs, and both **Uopj and LuBd** agreed the design is theoretically proven.
- **Technical Clarity and Execution:**  **MJAJ** and **Uopj** praised the work's execution, noting it is technically sound and conceptually clear.
- **SOTA Performance:** **Uopj** and **LuBd** acknowledged that HiDA-Net achieves state-of-the-art performance across multiple datasets.
- **Valuable Dataset Contribution:** The proposed **HiRes-50K** was recognized as a necessary contribution to the community, filling a gap in high-quality, high-resolution evaluation resources (**Uopj**, **L8Yu**, **LuBd**).

# **2. Key Concerns and Rebuttal Actions**

We have addressed all major and minor concerns through new experiments and detailed clarifications.

## A. Comparisons with New SOTA and Newer Generators

*Reviewer Uopj, LuBd*

- **Concern:** Requested comparisons with recent methods (SPAI[1], RINE[2]) and evaluation on newer generative models (e.g., FLUX, SD3).
- **Action:** We expanded our evaluations to include images from Midjourney v6.1, SD3, and FLUX. Under the SPAI testing protocol, our results confirm that HiDA-Net **significantly outperforms RINE** and achieves performance **comparable to SPAI**. Furthermore, on our more challenging HiRes-50K dataset (which is more representative of diverse, real-world internet distribution than benchmarks based on single-model generators), HiDA-Net demonstrated **a substantially larger performance advantage**. This highlights our model's competitiveness against the very latest baselines.

## B. Computational Complexity

*Reviewer Uopj, L8Yu*

- **Concern:** Questioned the inference cost of the transformer backbone and how it scales with high-resolution inputs.
- **Action:** We added a quantitative analysis (Section 6.4) and clarified that our tiling strategy ensures **FLOPs scale linearly with the number of image pixels** (total input size). This design avoids the quadratic complexity of standard full-image attention, making HiDA-Net highly efficient for high-resolution tasks.

## C. Dataset Construction and Value

*Reviewer Uopj, LuBd*

- **Concern:** Clarification on the dataset's value, source models, and resizing effects.
- **Action:** We clarified that HiRes-50K aggregates diverse, user-generated content from complex generation pipelines rather than fixed synthetic outputs.  We detailed our Lanczos downsampling protocol for real images to prevent upsampling artifacts. We further emphasized that HiRes-50K covers a unique range of **2MP to 64MP**, extending the evaluation boundary significantly beyond existing datasets (~1MP). The dataset underwent manual curation to filter out easily detectable samples, addressing the **current scarcity of high-resolution, difficult-to-detect forgery examples.**

## D. Methodological Ablations

*Reviewer L8Yu, MJAJ*

- **Concern:** Requested deeper analysis of the Dual-Path (Global vs. Local) design and robustness.
- **Action:** We added comprehensive ablation studies (Table 9) confirming that the Local path is critical for high-res performance while the Global path ensures semantic consistency. We also provided visual analyses of mixed-content (inpainted) images to demonstrate the model's precise localization capabilities.

We believe the revisions have fully addressed the reviewers' concerns. The addition of more comparisons, efficiency analysis, and expanded dataset details has strengthened the manuscript. We are confident that HiDA-Net and HiRes-50K will serve as valuable assets to the ICLR community.

[1] Karageorgiou, Dimitrios, et al. "Any-resolution ai-generated image detection by spectral learning," Proceedings of the Computer Vision and Pattern Recognition Conference. 2025.

[2] Koutlis, Christos, and Symeon Papadopoulos. "Leveraging representations from intermediate encoder-blocks for synthetic image detection," European Conference on Computer Vision, 2024.

---

### Meta-Review · Area_Chair_XeF9 · 2026-01-07

**Summary:**

This paper received mixed initial reviews. Reviewers mainly concern about the computation cost, comparisons with the baselines, diversity of the benchmark dataset, and limited visual analysis. AC reads the rebuttal and believes most of the concerns are well addressed in the authors' responses. To name a few, the authors made up the critical ablation studies on global/local branch, clarified the linearly scaling computation time w.r.t the image resolution, and explained the data sources and the diverse modern model results covered in the benchmark dataset. AC sees a clear improvement of the revised manuscript over the original version, and is glad to recommend the acceptance of this paper. More details are below.

**Reviewer Concerns:**

- Reviewer Uopj: the reviewer has concerns on the comparison with other SOTA baselines and the authors made up the experiments on SPAI and RINE, showing that the improvements on the performance is not marginal, especially on high-res images. Computational cost and data source in the set are also clarified well. AC believes the concerns of the reviewer have been well-addressed. However, the metadata from the benchmark dataset might still very important. It will be better if the authors can provide a subset of the dataset which contains detailed meta data, including the model, image region, and other information, for constructing a better benchmark.
- Reviewer L8Yu: questions regarding the architecture have been explained well by stating the light-weight adaptors. The major made-up experiments are the ablation on the local-global path design, and the new results demonstrate the effectiveness of the dual-path architecture. AC encourages the authors to provide more implementation details in the revised manuscript to address the reviewer's concerns.
- Reviewer LuBd: the reviewer mainly concerned about the generalization of the method in images from different models or having complicated degradation. The authors clarified the data sources, and provided additional experiments on the resizing, blur and noises. AC did not see outstanding issues.
- Reviewer MJAJ: as requested by the reviewer, the authors provided a visual analysis example of an inpainted image in the appendix. AC feels it will be helpful if more visual results are added. The reviewer's concern might remain outstanding. The experiments of tiling and rotation clarified the concerns well.

**Reviewer Scores:**

- Reviewer Uopj: the reviewer might remain the same score.
- Reviewer L8Yu: it is possible the reviewer will upgrade the scores, given the critical ablation is well added.
- Reviewer LuBd: the scores might be kept the same.
- Reviewer MJAJ: the newly added figures might not be sufficient enough (only an inpainted one is added). It should not be hard to have a bunch of examples, so AC guess the reviewer may remain the same score. But it is not a blocker for the acceptance.

---

### Decision · Program_Chairs · 2026-01-26

Accept (Poster)